# DISTANCE-BASED TREE-SLICED WASSERSTEIN DISTANCE

**Viet-Hoang Tran**[*]
Department of Mathematics
National University of Singapore
`hoang.tranviet@u.nus.edu`

**Khoi N.M. Nguyen**[*]
FPT Software AI Center
`khoinnm1@fpt.com`

**Trang Pham**
Qualcomm AI Research[◇]
`tranpham@qti.qualcomm.com`

**Thanh T. Chu**
Department of Computer Science
National University of Singapore
`thanh.chu@u.nus.edu`

**Tam Le**[†]
The Institute of Statistical Mathematics
& RIKEN AIP
`tam@ism.ac.jp`

**Tan M. Nguyen**[†]
Department of Mathematics
National University of Singapore
`tanmn@nus.edu.sg`

## ABSTRACT

To overcome computational challenges of Optimal Transport (OT), several variants of Sliced Wasserstein (SW) has been developed in the literature. These approaches exploit the closed-form expression of the univariate OT by projecting measures onto (one-dimensional) lines. However, projecting measures onto low-dimensional spaces can lead to a loss of topological information. Tree-Sliced Wasserstein distance on Systems of Lines (TSW-SL) has emerged as a promising alternative that replaces these lines with a more advanced structure called tree systems. The tree structures enhance the ability to capture topological information of the metric while preserving computational efficiency. However, at the core of TSW-SL, the splitting maps, which serve as the mechanism for pushing forward measures onto tree systems, focus solely on the position of the measure supports while disregarding the projecting domains. Moreover, the specific splitting map used in TSW-SL leads to a metric that is not invariant under Euclidean transformations, a typically expected property for OT on Euclidean space. In this work, we propose *a novel class of splitting maps* that generalizes the existing one studied in TSW-SL enabling the use of all positional information from input measures, resulting in a novel Distance-based Tree-Sliced Wasserstein (Db-TSW) distance. In addition, we introduce a simple tree sampling process better suited for Db-TSW, leading to an efficient GPU-friendly implementation for tree systems, similar to the original SW. We also provide a comprehensive theoretical analysis of proposed class of splitting maps to verify the injectivity of the corresponding Radon Transform, and demonstrate that Db-TSW is an Euclidean invariant metric. We empirically show that Db-TSW significantly improves accuracy compared to recent SW variants while maintaining low computational cost via a wide range of experiments on gradient flows, image style transfer, and generative models. The code is publicly available at `https://github.com/Fsoft-AIC/DbTSW`.

## 1 INTRODUCTION

Optimal transport (OT) (Villani, 2008; Peyré et al., 2019) is a framework designed to compare probability distributions by extending the concept of a ground cost metric, originally defined between the

---

[*] Co-first authors. [†] Co-last authors. [◇] Qualcomm Vietnam Company Limited. Correspondence to: hoang.tranviet@u.nus.edu & tanmn@nus.edu.sg

supports of input measures, to a metric between entire probability measures. OT has found utility across a diverse array of fields, including machine learning (Bunne et al., 2022; Hua et al., 2023; Nguyen et al., 2021b; Fan et al., 2022), data valuation (Just et al., 2023; Kessler et al., 2025), multimodal data analysis (Park et al., 2024; Luong et al., 2024), statistics (Mena & Niles-Weed, 2019; Weed & Berthet, 2019; Wang et al., 2022; Pham et al., 2024; Liu et al., 2022; Nguyen et al., 2022; Nietert et al., 2022), computer vision, and graphics (Lavenant et al., 2018; Nguyen et al., 2021a; Saleh et al., 2022; Solomon et al., 2015).

OT suffers from a computational burden due to its supercubic complexity with respect to the number of supports in the input measures (Peyré et al., 2019). To alleviate this issue, Sliced-Wasserstein (SW) (Rabin et al., 2011; Bonneel et al., 2015) leverages the closed-form solution of OT in the one-dimensional case to reduce computational demands by projecting the supports of the input measures onto random lines. The vanilla SW distance has been continuously developed and enhanced by refining existing components and introducing meaningful additions to achieve better performance. Examples include improvements in the sampling process (Nadjahi et al., 2021; Nguyen et al., 2024a), determining optimal projection lines (Deshpande et al., 2019), and modifying the projection mechanism (Kolouri et al., 2019; Bonet et al., 2023b).

**Related work.** Relying solely on one-dimensional projections can result in the loss of essential topological structures in high-dimensional data. To address this, an alternative approach has emerged that replaces these one-dimensional lines with different domains, applying to OT on Euclidean spaces (Alvarez-Melis et al., 2018; Paty & Cuturi, 2019; Niles-Weed & Rigollet, 2022), tree metric spaces (Indyk & Thaper, 2003; Le & Nguyen, 2021; Tran et al., 2025c;d), graph metric spaces (Le et al., 2022; 2023; 2024), spheres (Quellmalz et al., 2023; Bonet et al., 2023a; Tran et al., 2024b), and hyperbolic spaces (Bonet et al., 2023b). Specifically, Tran et al. (2025d) introduced an integration domain known as the tree system, which functions similarly to lines with a more advanced structure design. This approach is proven to capture the topological information better while maintaining the computational efficiency of the SW method. However, the proposed Tree-Sliced Wasserstein distance on Systems of Lines (TSW-SL) in (Tran et al., 2025d), derived from the tree system framework, fails to meet the Euclidean invariance property, which is typically expected for OT in Euclidean spaces (Alvarez-Melis et al., 2019). In this paper, we address these issues by generalizing the class of splitting maps introduced in (Tran et al., 2025d), simultaneously resolving the lack of invariance and the insufficient accounting for positional information. A new class of splitting maps may encounter challenges, such as verifying the injectivity of the associated Radon transform, resulting in a pseudo-metric on the space of measures.

**Contribution.** In summary, our contributions are three-fold:

1. We analyze the Euclidean invariance of 2-Wasserstein and Sliced $p$-Wasserstein distance between measures on Euclidean spaces and then discuss why Tree-Sliced Wasserstein distance on Systems of Lines fails to satisfy Euclidean invariance. To address this issue, we introduce a larger class of splitting maps that captures positional information from both points and tree systems, generalizing the previous class while incorporating an additional invariance property.

2. We introduce a novel variant of Radon Transform on Systems of Lines, developing a new class of invariant splitting maps that generalizes the previous version in (Tran et al., 2025d). By providing a comprehensive theoretical analysis with rigorous proofs, we demonstrate how our new class of invariant splitting maps ensures the injectivity of the Radon Transform.

3. We propose the novel Distance-based Tree-Sliced Wasserstein (Db-TSW) distance, which is an Euclidean invariant metric between measures. By analyzing the choice of splitting maps and tree systems in Db-TSW, we demonstrate that Db-TSW enables a highly parallelizable implementation, achieving an efficiency similar to that of the original SW.

**Organization.** The structure of the paper is as follows: Section 2 recalls variants of Wasserstein distance. Section 3 outlines the essential background of Tree-Sliced Wasserstein distance on Systems of Lines and discusses Euclidean invariance. Section 4 introduces a new class of splitting maps and discusses the corresponding Radon Transform. Section 5 proposes Distance-based Tree-Sliced Wasserstein (Db-TSW) distance and discusses the choices of components in Db-TSW. Finally, Section 6 evaluates Db-TSW performance. A complete theoretical framework of Db-TSW and supplemental materials are provided in the Appendix.

## 2 PRELIMINARIES

We review Wasserstein distance, Sliced Wasserstein (SW) distance, Wasserstein distance on metric spaces with tree metrics (TW), and Tree-Sliced Wasserstein on Systems of Lines (TSW-SL) distance.

**Wasserstein Distance.** Let $\mu$, $\nu$ be two probability distributions on $\mathbb{R}^d$. Let $\mathcal{P}(\mu,\nu)$ be the set of probability distributions $\pi$ on the product space $\mathbb{R}^d \times \mathbb{R}^d$ such that $\pi(A \times \mathbb{R}^d) = \mu(A)$ and $\pi(\mathbb{R}^d \times A) = \nu(A)$ for all measurable sets $A$. For $p \geqslant 1$, the $p$-Wasserstein distance $W_p$ (Villani, 2008) between $\mu$, $\nu$ is defined as:

$$W_p(\mu,\nu) = \inf_{\pi \in \mathcal{P}(\mu,\nu)} \left( \int_{\mathbb{R}^d \times \mathbb{R}^d} \|x - y\|_p^p \, d\pi(x,y) \right)^{\frac{1}{p}}. \tag{1}$$

**Sliced Wasserstein Distance.** The Sliced $p$-Wasserstein distance (SW) (Bonneel et al., 2015) between two probability distributions $\mu, \nu$ on $\mathbb{R}^d$ is defined by:

$$SW_p(\mu,\nu) := \left( \int_{\mathbb{S}^{d-1}} W_p^p(\mathcal{R}f_\mu(\cdot,\theta), \mathcal{R}f_\nu(\cdot,\theta)) \, d\sigma(\theta) \right)^{\frac{1}{p}}, \tag{2}$$

where $\sigma = \mathcal{U}(\mathbb{S}^{d-1})$ is the uniform distribution on the unit sphere $\mathbb{S}^{d-1}$, operator $\mathcal{R} : L^1(\mathbb{R}^d) \to L^1(\mathbb{R} \times \mathbb{S}^{d-1})$ is the Radon Transform (Helgason, 2011) defined by $\mathcal{R}f(t,\theta) = \int_{\mathbb{R}^d} f(x) \cdot \delta(t - \langle x, \theta \rangle) \, dx$, and $f_\mu, f_\nu$ are the probability density functions of $\mu, \nu$, respectively. The one-dimensional $p$-Wasserstein distance in Equation (2) has the closed-form $W_p^p(\theta \sharp \mu, \theta \sharp \nu) = \int_0^1 |F_{\mathcal{R}f_\mu(\cdot,\theta)}^{-1}(z) - F_{\mathcal{R}f_\nu(\cdot,\theta)}^{-1}(z)|^p dz$, where $F_{\mathcal{R}f_\mu(\cdot,\theta)}$ and $F_{\mathcal{R}f_\nu(\cdot,\theta)}$ are the cumulative distribution functions of $\mathcal{R}f_\mu(\cdot,\theta)$ and $\mathcal{R}f_\nu(\cdot,\theta)$, respectively. To approximate the intractable integral in Equation (2), Monte Carlo method is used as follows:

$$\widehat{SW}_p(\mu,\nu) = \left( \frac{1}{L} \sum_{l=1}^{L} W_p^p(\mathcal{R}f_\mu(\cdot,\theta_l), \mathcal{R}f_\nu(\cdot,\theta_l)) \right)^{\frac{1}{p}}, \tag{3}$$

where $\theta_1, \ldots, \theta_L$ are drawn independently from the uniform distributions on $\mathbb{S}^{d-1}$, i.e. $\mathcal{U}(\mathbb{S}^{d-1})$.

**Tree Wasserstein Distances.** Let $\mathcal{T}$ be a rooted tree (as a graph) with non-negative edge lengths, and the ground metric $d_\mathcal{T}$, i.e., the length of the unique path between two nodes. Given two probability distributions $\mu$ and $\nu$ supported on nodes of $\mathcal{T}$, the Wasserstein distance with ground metric $d_\mathcal{T}$ (TW) (Le et al., 2019) has closed-form as follows:

$$W_{d_\mathcal{T},1}(\mu,\nu) = \sum_{e \in \mathcal{T}} w_e \cdot \left| \mu(\Gamma(v_e)) - \nu(\Gamma(v_e)) \right|, \tag{4}$$

where $v_e$ is the endpoint of edge $e$ that is farther away from the tree root, $\Gamma(v_e)$ is the subtree of $\mathbb{T}$ rooted at $v_e$, and $w_e$ is the length of $e$.

**Tree-Sliced Wasserstein Distance on Systems of Lines.** Tree-Sliced Wasserstein distance on Systems of Lines (TSW-SL) (Tran et al., 2025d) is proposed as a combination between the projecting mechanism via Radon Transform in SW and the closed-form of Wasserstein distance with ground tree metrics in TW. In TSW-SL, tree systems, which are well-defined measure spaces and metric spaces with tree metric, are proposed to replace the role of directions in SW, and the corresponding variant of Radon Transform is also presented. Details of TSW-SL are outlined in Section 3.

## 3 THE ORIGINAL TSW-SL AND THE LACK OF $E(d)$-INVARIANCE

In this section, we outline the details of TSW-SL in (Tran et al., 2025d) and discuss the invariance under Euclidean transformations of TSW-SL and some variants of Wasserstein distance on Euclidean spaces.

### 3.1 REVIEW ON THE ORIGINAL TSW-SL

We recall the notion and theoretical results of the Radon Transform on Systems of Lines and the corresponding Tree-Sliced Wasserstein distance from (Tran et al., 2025d) with some modifications

on the notations. We start with a *function* $f \in L^1(\mathbb{R}^d)$. Usually, $d$ is the dimension of data, and $f$ is the distribution of data. A *line* in $\mathbb{R}^d$ is an element in $\mathbb{R}^d \times \mathbb{S}^{d-1}$, and a *system of $k$ lines* in $\mathbb{R}^d$ is an element of $(\mathbb{R}^d \times \mathbb{S}^{d-1})^k$. We denote a system of lines by $\mathcal{L}$, a line in $\mathcal{L}$ (also index) by $l$, and the space of all systems of $k$ lines by $\mathbb{L}_k^d$. The *ground set* of $\mathcal{L}$ is defined by:

$$\bar{\mathcal{L}} := \left\{ (x, l) \in \mathbb{R}^d \times \mathcal{L} : \ x = x_l + t_x \cdot \theta_l \text{ for an } t_x \in \mathbb{R} \right\},$$

where $x_l + t \cdot \theta_l$, $t \in \mathbb{R}$ is the *parameterization of $l$*. By abuse of notation, we sometimes index lines by $i = 1, \ldots, k$ and denote the line of index $i$ by $l_i$. The source and direction $l_i$ are denoted by $x_i$ and $\theta_i$, respectively. We also denote the collection of all systems of lines in $\mathbb{R}^d$ with $k$ lines by $\mathbb{L}_k^d$. A *tree system* is a system of lines $\mathcal{L}$ with an additional *tree structure* $\mathcal{T}$. It is a well-defined metric measure space and denoted by $(\mathcal{L}, \mathcal{T})$ or by $\mathcal{L}$ if the tree structure is not needed to be specific. A *space of trees* (i.e., collections of all tree systems with the same tree structure) is denoted by $\mathbb{T}$ with a probability distribution $\sigma$ on $\mathbb{T}$, which comes from the tree sampling process. For $\mathcal{L} \in \mathbb{L}_k^d$, the *space of Lebesgue integrable functions on $\mathcal{L}$* is

$$L^1(\mathcal{L}) = \left\{ f : \bar{\mathcal{L}} \to \mathbb{R} : \ \|f\|_{\mathcal{L}} = \sum_{l \in \mathcal{L}} \int_{\mathbb{R}} |f(t_x, l)| \, dt_x < \infty \right\}. \tag{5}$$

Given a splitting map $\alpha \in \mathcal{C}(\mathbb{R}^d, \Delta_{k-1})$, which is a continuous map from $\mathbb{R}^d$ to the $(k-1)$-dimensional standard simplex $\Delta_{k-1}$. For $f \in L^1(\mathbb{R}^d)$, we define

$$\mathcal{R}_{\mathcal{L}}^{\alpha} f : \quad \bar{\mathcal{L}} \quad \longrightarrow \mathbb{R} \tag{6}$$

$$(x, l) \longmapsto \int_{\mathbb{R}^d} f(y) \cdot \alpha(y)_l \cdot \delta \left( t_x - \langle y - x_l, \theta_l \rangle \right) \, dy. \tag{7}$$

The function $\mathcal{R}_{\mathcal{L}}^{\alpha} f$ is in $L^1(\mathcal{L})$. The operator

$$\mathcal{R}^{\alpha} : \ L^1(\mathbb{R}^d) \longrightarrow \prod_{\mathcal{L} \in \mathbb{L}_n^d} L^1(\mathcal{L})$$

$$f \longmapsto (\mathcal{R}_{\mathcal{L}}^{\alpha} f)_{\mathcal{L} \in \mathbb{L}_k^d}$$

is called the *Radon Transform on Systems of Lines*. This operator is *injective*. The *Tree-Sliced Wasserstein Distance on Systems of Lines* TSW-SL between $\mu, \nu \in \mathcal{P}(\mathbb{R}^d)$ is defined by

$$\text{TSW-SL}(\mu, \nu) = \int_{\mathbb{L}_k^d} \mathrm{W}_{d_{\mathcal{L}}, 1}(\mathcal{R}_{\mathcal{L}}^{\alpha} \mu, \mathcal{R}_{\mathcal{L}}^{\alpha} \nu) \, d\sigma(\mathcal{L}). \tag{8}$$

The TSW-SL is a metric on $\mathcal{P}(\mathbb{R}^d)$. Leveraging the closed-form expression of OT problems on metric spaces with tree metrics (Le et al., 2019) and the Monte Carlo method, TSW-SL in Equation (8) can be efficiently approximated by a closed-form expression.

*Remark* 1. The Radon Transform $\mathcal{R}^{\alpha}$ depends on the choice of $\alpha \in \mathcal{C}(\mathbb{R}^d, \Delta_{k-1})$. Intuitively, the splitting map $\alpha$ represents *how the mass at a specific point is distributed across all lines in a system of lines*. In the context of the original Radon Transform $\mathcal{R}$, only one line is involved, so $\alpha$ is simply the constant function 1. As a result, $\mathcal{R}^{\alpha}$ is a meaningful and nontrivial generalization of $\mathcal{R}$.

### 3.2 E($d$)-INVARIANCE IN OPTIMAL TRANSPORT ON EUCLIDEAN SPACES

Consider $\mathbb{R}^d$ with the Euclidean norm, i.e. $\| \cdot \|_2$, we consider some groups with group actions that preserve Euclidean norm and Euclidean distance between two points in $\mathbb{R}^d$. We then discuss E($d$)-invariance of some Wasserstein distance variants.

**Euclidean group E($d$) and its action on $\mathbb{R}^d$.** For $a \in \mathbb{R}^d$, the *translation corresponding to $a$* is the map $\mathbb{R}^d \to \mathbb{R}^d$ that $x \mapsto x + a$. The *translation group* T($d$) is the group of every translations in $\mathbb{R}^d$. Note that, T($d$) is isomorphic to the additive group $\mathbb{R}^d$. The *orthogonal group* O($d$) is the group of all linear transformations of $\mathbb{R}^d$ that preserve the Euclidean norm

$$\mathrm{O}(d) = \left\{ \text{linear transformation } f : \mathbb{R}^d \to \mathbb{R}^d : \ \|x\|_2 = \|f(x)\|_2 \text{ for all } x \in \mathbb{R}^d \right\}. \tag{9}$$

Note that O($d$) is isomorphic to the group of all orthogonal matrices

$$\mathrm{O}(d) = \left\{ Q \text{ is an } d \times d \text{ real matrix} : \ Q \cdot Q^{\top} = Q^{\top} \cdot Q = I_d \right\}. \tag{10}$$

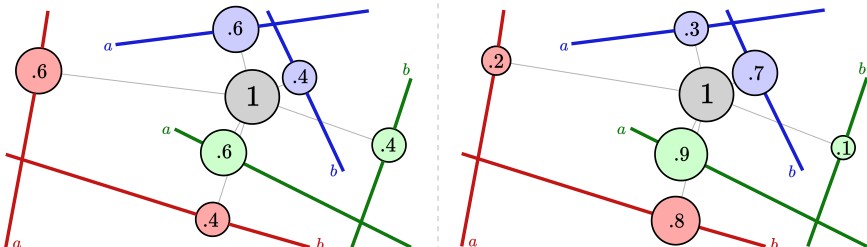

Figure 1: An illustration highlighting the distinction between the old and new Radon Transform on Systems of Lines, specifically focusing on two different definitions of splitting maps. *Left*: The old splitting map relies solely on the location of points, leading to the same distribution and independent of the position of the line systems. *Right*: The new splitting map considers the configuration of systems of lines, leading to varied mass distributions depending on each system.

The *Euclidean group* $E(d)$ is the group of all transformations of $\mathbb{R}^d$ that preserve the Euclidean distance between any two points. Formally, $E(d)$ is the semidirect product between $T(d)$ and $O(d)$, i.e., $E(d) \simeq T(d) \rtimes O(d)$. An element $g$ of $E(d)$ is denoted by a pair $g = (Q, a)$, where $a \in \mathbb{R}^d$ and $Q \in O(d)$. The action of $g$ on $\mathbb{R}^d$ is $y \mapsto gy = Q \cdot y + a$.

**Group actions of $E(d)$ on $\mathbb{L}_k^d$ and $\mathbb{T}$.** The canonical group action of $E(d)$ on $\mathbb{R}^d$ naturally induces a group action on the set of all lines in $\mathbb{R}^d$, i.e., $\mathbb{R}^d \times \mathbb{S}^{d-1}$. Given a line $l = (x, \theta) \in \mathbb{R}^d \times \mathbb{S}^{d-1}$ and $g = (Q, a) \in E(d)$, we define

$$gl := (Q \cdot x + a, Q \cdot \theta) \in \mathbb{R}^d \times \mathbb{S}^{d-1}. \tag{11}$$

For $\mathcal{L} = \left\{ l_i = (x_i, \theta_i) \right\}_{i=1}^k \in \mathbb{L}_k^d$, the action of $E(d)$ on $\mathbb{L}_k^d$ is similarly defined

$$g\mathcal{L} = \left\{ gl_i = (Q \cdot x_i + a, Q \cdot \theta_i) \right\}_{i=1}^k \in \mathbb{L}_k^d. \tag{12}$$

In (Tran et al., 2025d), a tree system is a system of lines with an additional tree structure, and by design, this tree structure is preserved under the action of $E(d)$. Thus, if $\mathcal{L} \in \mathbb{T}$ is a tree system, then $g\mathcal{L}$ is also a tree system. The group action of $E(d)$ on $\mathbb{L}_k^d$ induces a group action of $E(d)$ on $\mathbb{T}$.

$E(d)$**-equivariance in Optimal Transport.** $E(d)$-invariance is natural in the context of Optimal Transport on Euclidean space $\mathbb{R}^d$ since the distance between two measures should remain unchanged when a distance-preserving transformation is applied to the underlying space. In details, let $\mu \in \mathcal{P}(\mathbb{R}^n)$ be a measure on $\mathbb{R}^d$. For $g \in E(d)$, denote the pushforward of $\mu$ via $g : \mathbb{R}^d \to \mathbb{R}^d$ by $g\sharp\mu$. It canonically defines a group action of $E(d)$ on $\mathcal{P}(\mathbb{R}^n)$. We have the following result.

**Proposition 3.1.** *The* 2*-Wasserstein distance and the Sliced* $p$*-Wasserstein distance are* $E(d)$*-invariant. In other words, for every* $\mu, \nu \in \mathcal{P}(\mathbb{R}^d)$ *and* $g \in E(d)$*, we have*

$$W_2(\mu, \nu) = W_2(g\sharp\mu, g\sharp\nu) \ \text{ and } \ SW_p(\mu, \nu) = SW_p(g\sharp\mu, g\sharp\nu). \tag{13}$$

Moreover, for Sliced Wasserstein distance, we have the below result for each projecting direction.

**Proposition 3.2.** *Let* $g = (Q, a) \in E(d)$. *For every* $\mu, \nu \in \mathcal{P}(\mathbb{R}^d)$ *and* $\theta \in \mathbb{S}^{d-1}$*, we have*

$$W_p\left( \mathcal{R}f_\mu(\cdot, \theta), \mathcal{R}f_\nu(\cdot, \theta) \right) = W_p\left( \mathcal{R}f_{g\sharp\mu}(\cdot, Q\theta), \mathcal{R}f_{g\sharp\nu}(\cdot, Q\theta) \right) \tag{14}$$

The proofs for Proposition 3.1 and Proposition 3.2 can be found in Appendix B.1.

*Remark* 2. With the setting of TSW-SL, the $E(d)$-invariance and the similar property as Proposition 3.2 can not be derived for a general splitting map. Assumptions on invariance of splitting maps will be made to achieve invariance of TSW-SL.

## 4 RADON TRANSFORM WITH $E(d)$-INVARIANT SPLITTING MAPS

In (Tran et al., 2025d), the splitting maps are selected as continuous maps from $\mathbb{R}^d$ to $\Delta_{k-1}$, which means that it *depends solely on the position of points and ignores the tree systems*. It is intuitively

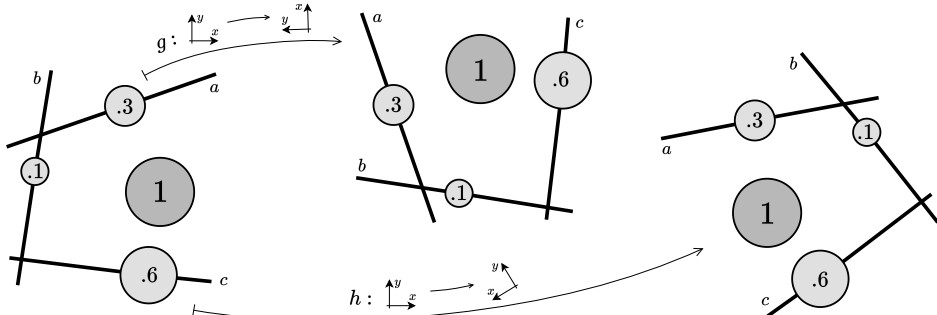

Figure 2: An illustration demonstrating E($d$)-invariance of splitting maps. Starting with a point and a system of lines, two Euclidean transformations are applied, resulting in two additional pairs of points and systems of lines. An E($d$)-invariant handles all three pairs identically, leading to the same mass distribution from the point to lines within each system.

better for splitting maps to account positional information from *both of points and tree systems*, than only from points. With this motivation, we introduce *a larger class of splitting maps with the corresponding Radon Transform* and then discuss *the injectivity of this Radon Transform variant*.

## 4.1 RADON TRANSFORM ON SYSTEMS OF LINES

Denote $\mathcal{C}(\mathbb{R}^d \times \mathbb{L}_k^d, \Delta_{k-1})$ as the space of continuous maps from $\mathbb{R}^d \times \mathbb{L}_k^d$ to $\Delta_{k-1}$ and refer to maps in $\mathcal{C}(\mathbb{R}^d \times \mathbb{L}_k^d, \Delta_{k-1})$ as *splitting maps*. Here, we use the same name as in (Tran et al., 2025d) since the new class of splitting maps contains the class of splitting maps in (Tran et al., 2025d).

$$\mathcal{C}(\mathbb{R}^d, \Delta_{k-1}) \xleftrightarrow{\ 1-1\ }$$
$$\{\alpha \in \mathcal{C}(\mathbb{R}^d \times \mathbb{L}_k^d, \Delta_{k-1}) : \ \alpha(x, \mathcal{L}) = \alpha(x, \mathcal{L}') \text{ for all } x \in \mathbb{R}^d \text{ and } \mathcal{L}, \mathcal{L}' \in \mathbb{L}_k^d\}.$$

Let $\mathcal{L}$ be a system of lines in $\mathbb{L}_n^d$ and $\alpha$ be a splitting map in $\mathcal{C}(\mathbb{R}^d \times \mathbb{L}_k^d, \Delta_{k-1})$, we define an operator associated to $\alpha$ that transforms a Lebesgue integrable functions on $\mathbb{R}^d$ to a Lebesgue integrable functions on $\mathcal{L}$. For $f \in L^1(\mathbb{R}^d)$, define

$$\mathcal{R}_{\mathcal{L}}^\alpha f : \ \bar{\mathcal{L}} \ \longrightarrow \mathbb{R} \tag{15}$$

$$(x, l) \longmapsto \int_{\mathbb{R}^d} f(y) \cdot \alpha(y, \mathcal{L})_l \cdot \delta\left(t_x - \langle y - x_l, \theta_l \rangle\right) \ dy, \tag{16}$$

where $\delta$ is the 1-dimensional Dirac delta function. We have $\mathcal{R}_{\mathcal{L}}^\alpha f \in L^1(\mathcal{L})$ for $f \in L^1(\mathbb{R}^d)$ and moreover $\|\mathcal{R}_{\mathcal{L}}^\alpha f\|_{\mathcal{L}} \leqslant \|f\|_1$. In other words, the linear operator $\mathcal{R}_{\mathcal{L}}^\alpha : L^1(\mathbb{R}^d) \to L^1(\mathcal{L})$ is well-defined. The proof of these properties can be found in Appendix B.2. We introduce a novel Radon Transform on Systems of Lines, which is a generalization of the variant in (Tran et al., 2025d).

**Definition 4.1** (Radon Transform on Systems of Lines). For $\alpha \in \mathcal{C}(\mathbb{R}^d \times \mathbb{L}_k^d, \Delta_{k-1})$, the operator

$$\mathcal{R}^\alpha : \ L^1(\mathbb{R}^d) \ \longrightarrow \ \prod_{\mathcal{L} \in \mathbb{L}_k^d} L^1(\mathcal{L}) \tag{17}$$

$$f \ \longmapsto \ (\mathcal{R}_{\mathcal{L}}^\alpha f)_{\mathcal{L} \in \mathbb{L}_k^d}, \tag{18}$$

is called the *Radon Transform on Systems of Lines*.

*Remark* 3. We use the same name and notation for $\mathcal{R}^\alpha$ as in (Tran et al., 2025d). Figure 1 emphasizes the difference between the old and new Radon Transform on Systems of Lines.

The injectivity of $\mathcal{R}^\alpha$ will be discussed in the next part. Surprisingly, the E($d$)-invariance of $\alpha$ is a sufficient condition for the injectivity of $\mathcal{R}^\alpha$.

## 4.2 E($d$)-INVARIANCE AND INJECTIVITY OF RADON TRANSFORM

Variants of Radon Transform usually require the transform to be injective. In the case of $\mathcal{R}^\alpha$, since we introduce a larger class of splitting maps $\alpha$, the proof for injectivity of the previous variant in

(Tran et al., 2025d) is not applicable for this new variant. We found that if $\alpha$ is E($d$)-invariant, then the injectivity of the Radon transform holds. We first have the definition of E($d$)-invariance of splitting maps.

**Definition 4.2.** A splitting map $\alpha$ in $\mathcal{C}(\mathbb{R}^d \times \mathbb{L}_k^d, \Delta_{k-1})$ is said to be E($d$)-invariant, if we have

$$\alpha(gy, g\mathcal{L}) = \alpha(y, \mathcal{L}) \tag{19}$$

for all $(y, \mathcal{L}) \in \mathbb{R}^d \times \mathbb{L}_k^d$ and $g \in \text{E}(d)$.

A visualization of E($d$)-invariant splitting maps is presented in Figure 2. Using this property of splitting maps, we get a result about injectivity of our Radon Transform variant.

**Theorem 4.3.** *For an* E($d$)*-invariant splitting map* $\alpha \in \mathcal{C}(\mathbb{R}^d \times \mathbb{L}_k^d, \Delta_{k-1})$, $\mathcal{R}^\alpha$ *is injective.*

The proof of Theorem 4.3 is presented in Appendix B.3.

*Remark* 4. Since the action of E($d$) on $\mathbb{R}^d$ is transitive, i.e. for $x, y \in \mathbb{R}^d$, there exists $g \in \text{E}(d)$ such that $gx = y$, so E($d$)-invariant splitting maps $\alpha \in \mathcal{C}(\mathbb{R}^d, \Delta_{k-1})$ must be constant. Thus, imposing invariance on previous splitting maps in (Tran et al., 2025d) significantly limits the class of maps.

Candidates for E($d$)-invariant splitting maps will be presented in the next section.

## 5 DISTANCE-BASED TREE-SLICED WASSERSTEIN DISTANCE

In this section, we present a novel Distance-based Tree-Sliced Wasserstein (Db-TSW) distance. Let consider the space of all tree systems of $k$ lines $\mathbb{T}$ and a distribution $\sigma$ on $\mathbb{T}$.

### 5.1 TREE-SLICED WASSERSTEIN DISTANCE WITH E($d$) SPLITTING MAP

For $\mu, \nu \in \mathcal{P}(\mathbb{R}^d)$, a tree system $\mathcal{L} \in \mathbb{T}$, and an E($d$)-invariant splitting map $\alpha \in \mathcal{C}(\mathbb{R}^d \times \mathbb{L}_k^d, \Delta_{k-1})$, by the Radon transform $\mathcal{R}_\mathcal{L}^\alpha$ in Definition 4.1, probability measures $\mu$ and $\nu$ are transformed to $\mathcal{R}_\mathcal{L}^\alpha \mu$ and $\mathcal{R}_\mathcal{L}^\alpha \nu$ in $\mathcal{P}(\mathcal{L})$. Notice that $\mathcal{L}$ is a metric space with tree metric $d_\mathcal{L}$ (Tran et al., 2025d), so we can compute Wasserstein distance $W_{d_\mathcal{L}, 1}(\mathcal{R}_\mathcal{L}^\alpha \mu, \mathcal{R}_\mathcal{L}^\alpha \nu)$ between $\mathcal{R}_\mathcal{L}^\alpha \mu$ and $\mathcal{R}_\mathcal{L}^\alpha \nu$ by Equation (4).

**Definition 5.1** (Distance-Based Tree-Sliced Wasserstein Distance)**.** The *Distance-based Tree-Sliced Wasserstein distance* between $\mu, \nu \in \mathcal{P}(\mathbb{R}^d)$ is defined by

$$\text{Db-TSW}(\mu, \nu) := \int_\mathbb{T} W_{d_\mathcal{L}, 1}(\mathcal{R}_\mathcal{L}^\alpha \mu, \mathcal{R}_\mathcal{L}^\alpha \nu) \, d\sigma(\mathcal{L}). \tag{20}$$

*Remark* 5. Note that, Db-TSW depends on the space of tree systems $\mathbb{T}$, the distribution $\sigma$ over $\mathbb{T}$, and the E($d$)-invariant splitting map $\alpha$. For simplicity, these are omitted from the notation. The choice of $\alpha$ will be discussed in the next part and is the reason for the name *Distance-based*.

The Monte Carlo method is utilized to estimate the intractable integral in Equation (20) as follows:

$$\widehat{\text{Db-TSW}}(\mu, \nu) = \frac{1}{L} \sum_{i=1}^L W_{d_{\mathcal{L}_i}, 1}(\mathcal{R}_{\mathcal{L}_i}^\alpha \mu, \mathcal{R}_{\mathcal{L}_i}^\alpha \nu), \tag{21}$$

where $\mathcal{L}_1, \dots, \mathcal{L}_L$ are drawn independently from the distribution $\sigma$ on $\mathbb{T}$. The Db-TSW distance is, indeed, a metric on $\mathcal{P}(\mathbb{R}^d)$. Moreover, Db-TSW is E($d$)-invariant.

**Theorem 5.2.** Db-TSW *is an* E($d$)*-invariant metric on* $\mathcal{P}(\mathbb{R}^d)$.

The proof of Theorem 5.2 is presented in Appendix B.4. We discuss the computation of Db-TSW in details in the next part.

### 5.2 COMPUTING DB-TSW

We construct the space of tree systems and E($d$)-invariant splitting maps that are suited for Db-TSW.

**Choices for the space of tree systems.** In (Tran et al., 2025d), the space of tree systems $\mathbb{T}$ used in computing Db-TSW is the collection of all tree systems with the tree structure is a chain of $k$

nodes as a graph. We discovered that this approach complicates the implementation of Db-TSW, resulting in a considerable increase in computation time. Here, we propose a *method for sampling tree systems* that simplifies the implementation in practice. Let $\mathbb{T}$ be the space of all tree systems, which consists of $k$ lines with the same source. In details, $\mathcal{L}$ in $\mathbb{T}$ can be presented as

$$\mathcal{L} = \big\{(x, \theta_1), (x, \theta_2), \ldots, (x, \theta_k)\big\}. \tag{22}$$

In other words, tree system $\mathcal{L}$ consists of $k$ concurrent lines with the same root. The distribution $\sigma$ on $\mathbb{T}$ is simply the joint distribution of $k + 1$ independent distributions, which are $\mu \in \mathcal{P}(\mathbb{R}^d)$ and $\mu_1, \mu_2, \ldots, \mu_k \in \mathcal{P}(\mathbb{S}^{d-1})$. In details, to sample a tree system $\mathcal{L} \in \mathbb{T}$, we

1. Sample $x \sim \mu$, where $\mu$ is a distribution on a bounded subset of $\mathbb{R}^d$, for instance, the uniform distribution on the $d$-dimensional hypercube $[-1, 1]^d$, i.e. $\mathcal{U}([-1, 1]^d)$; and,

2. For $i = 1, \ldots, k$, sample $\theta_i \sim \mu_i$, where $\mu_i$ is a distribution on $\mathbb{S}^{d-1}$, for instance, the uniform distribution $\mathcal{U}(\mathbb{S}^{d-1})$.

We also concentrate on a subcollection of $\mathbb{T}$, which includes tree systems that take into account the angles between pairs of lines. Assume that $k \leqslant d$, denote $\mathbb{T}^{\perp}$ as the space of tree systems $\mathcal{L}$ consists of $k$ concurrent lines with the same root, and these lines are mutually orthogonal, i.e. $\mathcal{L} = \{(x, \theta_1), (x, \theta_2), \ldots, (x, \theta_k)\}$ where $\langle \theta_i, \theta_j \rangle = 0$ for all $1 \leqslant i < j \leqslant k$. The intuitive motivation for this choice is to ensure that the sampled tree systems do not include lines with similar directions. *Remark* 6. The condition $k \leqslant d$ arises because $\{\theta_1, \ldots, \theta_k\}$ forms a subset of an orthogonal basis of $\mathbb{R}^d$. In practice, this has little impact, as the dimension $d$ is typically large, while the number of directions $k$ is selected to be small.

**Choices for $\mathrm{E}(d)$-invariant splitting maps.** Since the action of $\mathrm{E}(d)$ preserves the distance between two points in $\mathbb{R}^d$, it also preserves the distance between a point and a line in $\mathbb{R}^d$. For $x \in \mathbb{R}^d$ and $\mathcal{L} \in \mathbb{L}_k^d$, we denote the distance between $x$ and line $l$ of $\mathcal{L}$ as $d(x, \mathcal{L})_l$ given by:

$$d(x, \mathcal{L})_l = \inf_{y \in l} \|x - y\|_2. \tag{23}$$

We have $d(x, \mathcal{L})_l$ is $\mathrm{E}(d)$-invariant. As a result, a splitting map $\alpha \colon \mathbb{R}^d \to \Delta_{k-1}$ that is in the form

$$\alpha(x, \mathcal{L})_l = \beta\Big(\{d(x, \mathcal{L})_l\}_{l \in \mathcal{L}}\Big), \tag{24}$$

where $\beta$ is an arbitrary map $\mathbb{R}^k \to \Delta_{k-1}$, is $\mathrm{E}(d)$-invariant. In practice, we empirically observed that choosing $\beta$ to be the $\mathrm{softmax}$ function together with scaling by a scalar works well in applications. In details, we choose $\alpha$ as follows

$$\alpha(x, \mathcal{L})_l = \mathrm{softmax}\Big(\{\delta \cdot d(x, \mathcal{L})_l\}_{l \in \mathcal{L}}\Big) \tag{25}$$

Here, $\delta \in \mathbb{R}$ is considered as to be a tuning parameter.[1] Intuition behind this choice of $\alpha$ is that it reflects the proximity of points to lines in tree systems. As $|\delta|$ grows, the output of $\alpha$ tends to become more sparse, emphasizing the importance of each line in the tree system relative to a specific point. We summarize the computation of Db-TSW by Algorithm 1.
*Remark* 7. The above construction of $\alpha$ mainly bases on the distance between points and lines in tree systems. That is the reason for the name *Distance-based* Tree-Sliced Wasserstein.

## 6 EXPERIMENTAL RESULTS

In this section, we experimentally demonstrate the advantages of our Db-TSW methods over traditional SW distance and its variants across three key domains: unconditional image synthesis, gradient flows, and color transfer. Detailed experimental settings are provided in Appendix §C.1. Our evaluation aims to establish that: (i) Db-TSW and Db-TSW$^{\perp}$[2] consistently enhance performance across various generative and optimization tasks; (ii) Db-TSW$^{\perp}$ significantly outperforms not only baselines but also Db-TSW in all tasks, highlighting its superiority; and (iii) Db-TSW and Db-TSW$^{\perp}$ is universal applicable and can be seamlessly integrated into any Optimal Transport task.

---

[1] We abuse the notation $\delta \in \mathbb{R}$ for the splitting map function $\alpha$.
[2] $\perp$ stands for using orthogonal directions $\theta$.

Table 1: Results of different DDGAN variants for unconditional generation on CIFAR-10.

| Model | FID ↓ | Time/Epoch(s) ↓ |
|---|---|---|
| DDGAN (Xiao et al. (2022)) | 3.64 | 136 |
| SW-DD (Nguyen et al. (2024b)) | 2.90 | 140 |
| DSW-DD (Nguyen et al. (2024b)) | 2.88 | 1059 |
| EBSW-DD (Nguyen et al. (2024b)) | 2.87 | 145 |
| RPSW-DD (Nguyen et al. (2024b)) | 2.82 | 159 |
| IWRPSW-DD (Nguyen et al. (2024b)) | 2.70 | 152 |
| TSW-SL-DD (Tran et al. (2025d)) | 2.83 | 163 |
| Db-TSW-DD (Ours) | 2.60 | 160 |
| Db-TSW-DD$^\perp$ (Ours) | **2.525** | 162 |

We maintain a consistent total number of sampled lines across all methods to ensure fair comparison. Db-TSW and Db-TSW$^\perp$ offer two important advantages. First, these methods contain richer information compared to SW for the same number of data points, as they incorporate positional information from both points and tree systems. This allows for more effective sampling, specifically by sampling trees located around the support of target distributions (ideally near their means). Second, the root-concurrent tree system design enables an algorithm with linear runtime and high parallelizability, keeping the wall clock time of Db-TSW and Db-TSW$^\perp$ approximately equal to vanilla SW and surpassing some SW variants.

## 6.1 DIFFUSION MODELS

This experiment explores the efficacy of denoising diffusion models for unconditional image synthesis. We employ a variant of the Denoising Diffusion Generative Adversarial Network (DDGAN) (Xiao et al., 2022), as introduced by Nguyen et al. (2024b), which incorporates a Wasserstein distance within the Augmented Generalized Mini-batch Energy (AGME) loss function. A detailed explanation of this Optimal Transport-based DDGAN is provided in Appendix §C.2. We evaluate our proposed methods, Db-TSW-DD and Db-TSW-DD$^\perp$, against DDGAN and various OT-based DDGAN variants, as enumerated in Table 1. All models undergo training for 1800 epochs on the CIFAR10 dataset (Krizhevsky et al., 2009). For vanilla SW and its variants, we adopt the parameters from Nguyen et al. (2024b) and set $L = 10000$. For Db-TSW-DD$^\perp$ and Db-TSW-DD models, we set $L = 2500, k = 4, \delta = 10$. Tree sampling is conducted from a Gaussian distribution $\mathcal{N}(m_t, \sigma^2 I)$, where $m_t$ represents the mean of all training samples and $\sigma = 0.1$. Table 1 presents the Fréchet Inception Distance (FID) scores and per-epoch training times on an Nvidia V100 GPU for our methods and the baselines. Lower FID scores indicate superior model performance. The results demonstrate that both Db-TSW-DD$^\perp$ and Db-TSW-DD achieve notable improvements in FID compared to all baselines. Specifically, they outperform the current state-of-the-art OT-based DDGAN, IWRPSW-DD (Nguyen et al., 2024b), by margins of 0.175 and 0.1, respectively. Importantly, our methods maintain computational efficiency, with only a modest 6.5% increase in training time compared to the current state-of-the-art.

## 6.2 GRADIENT FLOWS

The gradient flow task aims to minimize the distance between source and target distributions using gradient descent process. The objective is to iteratively reduce this distance by optimizing the equation $\partial_t \mu_t = -\nabla \mathcal{D}(\mu_t, \nu)$ with the initial condition $\mu_0 = \mathcal{N}(0, 1)$, where $\mu_t$ represents the source distribution at time $t$, $-\partial_t \mu_t$ denotes the change in the source distribution over time, and $\nabla \mathcal{D}(\mu_t, \nu)$ is the gradient of $\mathcal{D}$ with respect to $\mu_t$. Here, $\mathcal{D}$ refers to our proposed distance metric and the SW baselines. We evaluate our approach Db-TSW$^\perp$, Db-TSW and several established techniques, including vanilla SW Bonneel et al. (2015), MaxSW (Deshpande et al., 2019), LCVSW (Nguyen & Ho, 2024), SWGG (Mahey et al., 2023), and TSW-SL (Tran et al., 2025d), using the *Swiss Roll* and *Gaussian 20d* datasets.

To assess the effectiveness of these methods, we employ the Wasserstein distance to quantify average Wasserstein distances between the source and target distributions across 10 runs at iterations 500, 1000, 1500, 2000 and 2500. The results demonstrated in Table 2 show Db-TSW and Db-TSW$^\perp$ record the best result on almost all steps in Swiss Roll dataset, with a 3.78e-8 $W_2$ on the last step

Table 2: Average Wasserstein distance between source and target distributions of 10 runs on Swiss Roll and Gaussian 20d datasets. All methods use 100 projecting directions.

| Methods | Swiss Roll | | | | | | Gaussian 20d | | | | | |
| | Iteration | | | | | Time/Iter($s$) | Iteration | | | | | Time/Iter($s$) |
| | 500 | 1000 | 1500 | 2000 | 2500 | | 500 | 1000 | 1500 | 2000 | 2500 | |
|---|---|---|---|---|---|---|---|---|---|---|---|---|
| SW | 5.73e-3 | 2.04e-3 | 1.23e-3 | 1.11e-3 | 1.05e-3 | 0.009 | 18.24 | 16.48 | 14.66 | 12.60 | 10.30 | 0.006 |
| MaxSW | 2.47e-2 | 1.03e-2 | 6.10e-3 | 4.47e-3 | 3.45e-3 | 2.46 | 13.24 | 13.71 | 13.46 | 12.71 | 11.83 | 2.38 |
| SWGG | 3.84e-2 | 1.53e-2 | 1.02e-2 | 4.49e-3 | 3.57e-5 | 0.011 | 8.51 | 8.71 | 7.71 | 8.48 | 6.45 | 0.009 |
| LCVSW | 7.28e-3 | 1.40e-3 | 1.38e-3 | 1.38e-3 | 1.36e-3 | 0.010 | 17.26 | 14.70 | 12.00 | 9.04 | 6.15 | 0.009 |
| TSW-SL | 9.41e-3 | 2.03e-7 | 9.63e-8 | 4.44e-8 | 3.65e-8 | 0.014 | 3.13 | 9.67e-3 | 6.81e-3 | 6.15e-3 | 5.71e-3 | 0.010 |
| Db-TSW | 5.47e-3 | 8.04e-8 | 5.29e-8 | 3.92e-8 | 3.01e-8 | 0.006 | 3.27 | 1.12e-2 | 7.21e-3 | 5.63e-3 | 4.60e-3 | 0.006 |
| Db-TSW$^\perp$ | 7.55e-3 | 2.65e-7 | 4.90e-8 | 4.18e-8 | 3.78e-8 | 0.009 | 2.46 | 7.96e-3 | 6.11e-3 | 5.22e-3 | 5.03e-3 | 0.009 |

in comparison with 1.05e-3 of vanilla SW and 3.57e-5 of SWGG. A similar trend is observed in the Gaussian 20d dataset, with Db-TSW$^\perp$ and Db-TSW consistently achieving the second best and best Wasserstein distances respectively, outperforming the next best method SWGG by three orders of magnitude (5.03e-3 and 4.60e-3 vs. 7.34) and vanilla SW by nearly four orders of magnitude (10.30) at the final iteration. Notably, Db-TSW and Db-TSW$^\perp$ maintain competitive computational efficiency, with runtimes equal to good LCVSW variant (0.009 seconds per iteration) while boosting the performance significantly.

## 6.3 COLOR TRANSFER

The color transfer task involves transferring color from a source image to a target image. Considering source and target color palettes as $X$ and $Y$, we define the curve $\dot{Z}(t) = -n\nabla_{Z(t)}[\mathcal{D}(P_{Z(t)}, P_Y)]$ where $P_X$ and $P_Y$ are empirical distributions over $X$ and $Y$ and $\mathcal{D}$ are customizable Wasserstein distance. Here, the curve starts from $Z(0) = X$ and ends at $Y$. We iterate along this curve to perform the transfer between the probability distributions $P_X$ and $P_Y$. We evaluate our method against SW, TSW-SL, MaxSW, and various Quasi-Sliced Wasserstein (QSW) variants (Nguyen et al., 2024a). We follow the settings used in (Nguyen et al., 2024a) for experiment settings, detailed in Appendix §C.4. We set $L = 33, k = 3$ for Db-TSW and Db-TSW$^\perp$ and $L = 100$ for all baselines.

In Figure 5 (Appendix), we compare the Wasserstein distances and visualizations of various color transfer methods. Notably, Db-TSW$^\perp$ and Db-TSW achieve near-best performance (0.12 and 0.21 respectively) without randomization overhead of top-performing R-methods like RRQDSW and RQDSW (both at 0.08) and significantly outperforming traditional approaches such as SW and MaxSW (both at 9.58).

## 7 CONCLUSION

The paper presents the Distance-based Tree-Sliced Wasserstein (Db-TSW) distance for comparing measures in Euclidean spaces. Built on the Tree-Sliced Wasserstein distance on Systems of Lines (TSW-SL) framework, Db-TSW enhances the class of splitting maps that are central to TSW-SL. This new class of splitting maps in Db-TSW resolves several challenges in TSW-SL, such as the lack of Euclidean invariance, insufficient consideration of positional information, and slower computational performance. To address the theoretical gaps regarding these new maps, we provide a comprehensive theoretical framework with rigorous proofs for key properties of Db-TSW. Our experiments show that Db-TSW outperforms recent Sliced Wasserstein (SW) variants on a wide range of task, such as gradient flows, or generative models as GAN or diffusion models. In comparison to these SW variants, Db-TSW primarily focuses on refining the integration domain using tree systems and the associated measure projection mechanism through splitting maps, which are not existed in SW framework. Therefore, adapting to Db-TSW several techniques used in proving the SW variant is expected to be beneficial for future research. Additionally, the System of Lines for TSW provides advanced geometric structure to go beyond the lines for SW, which is a promising research direction for further investigation, e.g., the concurrent TSW-SL on the sphere (Tran et al., 2025b).

ACKNOWLEDGMENTS

This research / project is supported by the National Research Foundation Singapore under the AI Singapore Programme (AISG Award No: AISG2-TC-2023-012-SGIL). This research / project is supported by the Ministry of Education, Singapore, under the Academic Research Fund Tier 1 (FY2023) (A-8002040-00-00, A-8002039-00-00). This research / project is also supported by the NUS Presidential Young Professorship Award (A-0009807-01-00) and the NUS Artificial Intelligence Institute–Seed Funding (A-8003062-00-00).

We thank the area chairs, anonymous reviewers for their comments. TL acknowledges the support of JSPS KAKENHI Grant number 23K11243, and Mitsui Knowledge Industry Co., Ltd. grant.

**Ethics Statement.** Given the nature of the work, we do not foresee any negative societal and ethical impacts of our work.

**Reproducibility Statement.** Source codes for our experiments are provided in the supplementary materials of the paper. The details of our experimental settings and computational infrastructure are given in Section 6 and the Appendix. All datasets that we used in the paper are published, and they are easy to access in the Internet.

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

NOTATION

| | |
|---|---|
| $\mathbb{R}^d$ | $d$-dimensional Euclidean space |
| $\|\cdot\|_2$ | Euclidean norm |
| $\langle\cdot,\cdot\rangle$ | standard dot product |
| $\mathbb{S}^{d-1}$ | $(d-1)$-dimensional hypersphere |
| $\theta$ | unit vector |
| $\sqcup$ | disjoint union |
| $L^1(X)$ | space of Lebesgue integrable functions on $X$ |
| $\mathcal{P}(X)$ | space of probability distributions on $X$ |
| $\mu, \nu$ | measures |
| $\delta(\cdot)$ | 1-dimensional Dirac delta function |
| $\mathcal{U}(\mathbb{S}^{d-1})$ | uniform distribution on $\mathbb{S}^{d-1}$ |
| $\sharp$ | pushforward (measure) |
| $\mathcal{C}(X,Y)$ | space of continuous maps from $X$ to $Y$ |
| $d(\cdot,\cdot)$ | metric in metric space |
| $\mathrm{T}(d)$ | translation group of order $d$ |
| $\mathrm{O}(d)$ | orthogonal group of order $d$ |
| $\mathrm{E}(d)$ | Euclidean group of order $d$ |
| $g$ | element of group |
| $\mathrm{W}_p$ | $p$-Wasserstein distance |
| $\mathrm{SW}_p$ | Sliced $p$-Wasserstein distance |
| $\Gamma$ | (rooted) subtree |
| $e$ | edge in graph |
| $w_e$ | weight of edge in graph |
| $l$ | line, index of line |
| $\mathcal{L}$ | system of lines, tree system |
| $\bar{\mathcal{L}}$ | ground set of system of lines, tree system |
| $\Omega_{\mathcal{L}}$ | topological space of system of lines |
| $\mathbb{L}_k^d$ | space of symtems of $k$ lines in $\mathbb{R}^d$ |
| $\mathcal{T}$ | tree structure in system of lines |
| $L$ | number of tree systems |
| $k$ | number of lines in a system of lines or a tree system |
| $\mathcal{R}$ | original Radon Transform |
| $\mathcal{R}^\alpha$ | Radon Transform on Systems of Lines |
| $\Delta_{k-1}$ | $(k-1)$-dimensional standard simplex |
| $\alpha$ | splitting map |
| $\delta$ | tuning parameter in splitting maps |
| $\mathbb{T}$ | space of tree systems |
| $\mathbb{T}^\perp$ | space of orthogonal tree systems |
| $\sigma$ | distribution on space of tree systems |

# Supplement for
# "Distance-Based Tree-Sliced Wasserstein Distance"

## Table of Contents

## A  BACKGROUND FOR TREE-SLICED WASSERSTEIN DISTANCE ON SYSTEMS OF LINES

This section provides background for Tree-Sliced Wasserstein distance on Systems of Lines. For completeness, we recall essential definitions and theoretical results. Proofs and further details can be found in (Tran et al., 2025d).

### A.1  TREE SYSTEM

A *line* in $\mathbb{R}^d$ is determined by a pair $(x, \theta) \in \mathbb{R}^d \times \mathbb{S}^{d-1}$ and is parameterized as $x + t \cdot \theta$, where $t \in \mathbb{R}$. A line in $\mathbb{R}^d$ is denoted or indexed by $l = (x_l, \theta_l) \in \mathbb{R}^d \times \mathbb{S}^{d-1}$. Here, $x_l$ and $\theta_l$ are referred to as the *source* and *direction* of $l$, respectively. For $k \geqslant 1$, a *system of $k$ lines in $\mathbb{R}^d$* is a set of $k$ lines. We denote $(\mathbb{R}^d \times \mathbb{S}^{d-1})^k$ by $\mathbb{L}_k^d$, which represents the *space of systems of $k$ lines in $\mathbb{R}^d$*, and an element of $\mathbb{L}_k^d$ is typically denoted by $\mathcal{L}$. The system $\mathcal{L}$ is said to be connected if the points on the lines form a connected subset of $\mathbb{R}^d$. By *removing* some intersections between lines, we obtain a tree system $\mathcal{L}$, where there is a unique path between any two points of $\mathcal{L}$. For a quick visualization of tree systems, kindly refer to Figure 3.

*Remark* 8. The term *tree system* is used because there is a unique path between any two points, analogous to the definition of trees in graph theory.

By identifying all remaining intersections, together with the notions of disjoint union topology and quotient topology (Hatcher, 2005), we obtain a topology on a tree system as the gluing of copies of $\mathbb{R}$. Analyzing this topology, we find that tree systems are metric spaces with a tree metric.

## A.2 SAMPLING PROCESS OF TREE SYSTEMS

The space of tree systems exhibits significant diversity due to the various possible choices for tree structures (as graphs). (Tran et al., 2025d) outlines a comprehensive approach applicable to general tree structures, and focuses on the implementation of chain-like tree structures. The process of sampling tree systems based on this chain-like structure is as follows:

*Step* 1. Sample $x_1 \sim \mu_1$ and $\theta_1 \sim \nu_1$ for $\nu_1 \in \mathcal{P}(\mathbb{S}^{d-1})$ and $\mu_1 \in \mathcal{P}(\mathbb{R}^d)$.

*Step* $i$. Sample $x_i = x_{i-1} + t_i \cdot \theta_{i-1}$ where $t_i \sim \mu_i$ and $\theta_i \sim \nu_i$ for $\mu_i \in \mathcal{P}(\mathbb{R})$ and $\nu_i \in \mathcal{P}(\mathbb{S}^{d-1})$.

We assume all $\mu$'s and $\nu$'s are independent, and let:

1. $\mu_1$ to be a distribution on a bounded subset of $\mathbb{R}^d$, for instance, the uniform distribution on the $d$-dimensional cube $[-1,1]^d$, i.e. $\mathcal{U}([-1,1]^d)$;

2. $\mu_i$ for $i > 1$ to be a distribution on a bounded subset of $\mathbb{R}$, for example, the uniform distribution on the interval $[-1,1]$, i.e. $\mathcal{U}([-1,1])$;

3. $\theta_n$ to be a distribution on $\mathbb{S}^{d-1}$, for example, the uniform distribution $\mathcal{U}(\mathbb{S}^{d-1})$.

We derive a distribution, denoted by $\sigma$, over the space of all tree systems that can be sampled in this way, denoted by $\mathbb{T}$. The tree system shown in Figure 3 is an example of a tree system with a chain-like structure.

## A.3 RADON TRANSFORM ON SYSTEMS OF LINES

Denote $L^1(\mathbb{R}^d)$ as the space of Lebesgue integrable functions on $\mathbb{R}^d$ with norm $\|\cdot\|_1$. Let $\mathcal{L} \in \mathbb{L}_k^d$ be a system of $k$ lines. A *Lebesgue integrable function on* $\mathcal{L}$ is a function $f \colon \bar{\mathcal{L}} \to \mathbb{R}$ such that:

$$\|f\|_{\mathcal{L}} := \sum_{l \in \mathcal{L}} \int_{\mathbb{R}} |f(t_x, l)|\, dt_x < \infty. \tag{26}$$

Denote $L^1(\mathcal{L})$ as the *space of Lebesgue integrable functions on* $\mathcal{L}$.

Denote $\mathcal{C}(\mathbb{R}^d, \Delta_{k-1})$ as the space of continuous maps from $\mathbb{R}^d$ to the $(k-1)$-dimensional standard simplex $\Delta_{k-1} = \{(a_l)_{l \in \mathcal{L}} : a_l \geqslant 0 \text{ and } \sum_{l \in \mathcal{L}} a_l = 1\} \subset \mathbb{R}^k$. A map in $\mathcal{C}(\mathbb{R}^d, \Delta_{k-1})$ is called a *splitting map*. Let $\mathcal{L} \in \mathbb{L}_k^d$, $\alpha \in \mathcal{C}(\mathbb{R}^d, \Delta_{k-1})$, we define a linear operator $\mathcal{R}_{\mathcal{L}}^\alpha$ that transforms a function in $L^1(\mathbb{R}^d)$ to a function in $L^1(\mathcal{L})$. For $f \in L^1(\mathbb{R}^d)$, define:

$$\mathcal{R}_{\mathcal{L}}^\alpha f : \quad \bar{\mathcal{L}} \longrightarrow \mathbb{R}$$
$$(x, l) \longmapsto \int_{\mathbb{R}^d} f(y) \cdot \alpha(y)_l \cdot \delta\left(t_x - \langle y - x_l, \theta_l \rangle\right)\, dy \tag{27}$$

The (old) Radon Transform for Systems of Lines $\mathcal{R}^\alpha$ in (Tran et al., 2025d) is defined as follows:

$$\mathcal{R}^\alpha : \quad L^1(\mathbb{R}^d) \longrightarrow \prod_{\mathcal{L} \in \mathbb{L}_k^d} L^1(\mathcal{L})$$
$$f \longmapsto (\mathcal{R}_{\mathcal{L}}^\alpha f)_{\mathcal{L} \in \mathbb{L}_k^d}.$$

As shown in (Tran et al., 2025d), $\mathcal{R}^\alpha$ is injective for all splitting maps $\alpha \in \mathcal{C}(\mathbb{R}^d, \Delta_{k-1})$.

## A.4 TREE-SLICED WASSERSTEIN DISTANCE ON SYSTEMS OF LINES (TSW-SL)

Given the space of tree systems $\mathbb{T}$, distribution $\sigma$ on $\mathbb{T}$, $\alpha \in \mathcal{C}(\mathbb{R}^d, \Delta_{k-1})$, the *Tree-Sliced Wasserstein distance on Systems of Lines* (Tran et al., 2025d) between $\mu, \nu$ in $\mathcal{P}(\mathbb{R}^d)$ is defined by

$$\text{TSW-SL}(\mu, \nu) = \int_{\mathbb{T}} W_{d_{\mathcal{L}}, 1}(\mathcal{R}_{\mathcal{L}}^\alpha \mu, \mathcal{R}_{\mathcal{L}}^\alpha \nu)\, d\sigma(\mathcal{L}). \tag{28}$$

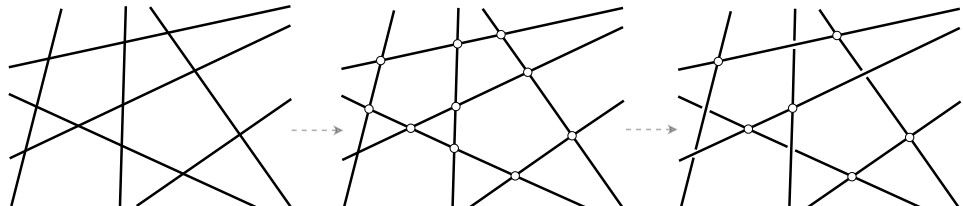

Figure 3: An illustration of constructing a tree system: Starting with a bunch of lines with no structure (left), we consider the intersections of all pairs of lines (middle), then removing some of the intersections to obtain a tree system (Right). There exists a unique path between any two points, since we only allows the pass through the remained intersections.

TSW-SL is indeed a metric on $\mathcal{P}(\mathbb{R}^d)$. The proof in (Tran et al., 2025d) primarily relies on the injectivity of the (old) Radon Transform on Systems of Lines $\mathcal{R}^\alpha$.

# B THEORETICAL PROOFS

## B.1 PROOF OF PROPOSITION 3.1 AND PROPOSITION 3.2

**Invariance and Equivariance Properties in Machine Learning.** Equivariant networks (Cohen & Welling, 2016) enhance generalization and improve sample efficiency by embedding task symmetries into the model architecture. They have shown considerable success in a range of domains such as trajectory prediction (Walters et al., 2020), robotics (Simeonov et al., 2022), graph-based models (Satorras et al., 2021; Tran et al., 2024a), functional networks (Tran et al., 2025a; 2024c; Vo et al., 2024), etc. Utilizing equivariance has been found to boost performance, increase data efficiency, and enhance robustness against out-of-domain generalization.

*Proof.* For 2-Wasserstein distance, recall that, for $\mu, \nu \in \mathcal{P}(\mathbb{R}^d)$, we have

$$\mathrm{W}_2(\mu,\nu) = \inf_{\pi \in \mathcal{P}(\mu,\nu)} \left( \int_{\mathbb{R}^d \times \mathbb{R}^d} \|x - y\|_2^2 \, d\pi(x,y) \right)^{\frac{1}{2}}. \tag{29}$$

We need to show that: for $g \in \mathrm{E}(d)$,

$$\mathrm{W}_2(\mu,\nu) = \mathrm{W}_2(g\sharp\mu, g\sharp\nu). \tag{30}$$

For $g = (Q, a)$, the map $g : \mathbb{R}^d \to \mathbb{R}^d$ is an affine map that preserves Euclidean norm $\|\cdot\|_2$ and $\det(Q) = 1$. By applying change of variable, we have

$$\mathrm{W}_2(g\sharp\mu, g\sharp\nu) = \inf_{\pi \in \mathcal{P}(g\sharp\mu, g\sharp\nu)} \left( \int_{\mathbb{R}^d \times \mathbb{R}^d} \|x - y\|_2^2 \, d\pi(x,y) \right)^{\frac{1}{2}} \tag{31}$$

$$= \inf_{\pi \in \mathcal{P}(\mu,\nu)} \left( \int_{\mathbb{R}^d \times \mathbb{R}^d} \|x - y\|_2^2 \, d(g\sharp\pi)(x,y) \right)^{\frac{1}{2}} \tag{32}$$

$$= \inf_{\pi \in \mathcal{P}(\mu,\nu)} \left( \int_{\mathbb{R}^d \times \mathbb{R}^d} \|x - y\|_2^2 \, d\pi(g^{-1}x, g^{-1}y) \right)^{\frac{1}{2}} \tag{33}$$

$$= \inf_{\pi \in \mathcal{P}(\mu,\nu)} \left( \int_{\mathbb{R}^d \times \mathbb{R}^d} \|gx - gy\|_2^2 \, d\pi(x,y) \right)^{\frac{1}{2}} \tag{34}$$

$$= \inf_{\pi \in \mathcal{P}(\mu,\nu)} \left( \int_{\mathbb{R}^d \times \mathbb{R}^d} \|x - y\|_2^2 \, d\pi(x,y) \right)^{\frac{1}{2}} \tag{35}$$

$$= \mathrm{W}_2(\mu,\nu). \tag{36}$$

We finish the proof for 2-Wasserstein distance. For Sliced $p$-Wasserstein distance, we first show that, for $g = (Q, a) \in \mathrm{E}(d)$, $\mu, \nu \in \mathcal{P}(\mathbb{R}^d)$ and $\theta \in \mathbb{S}^{d-1}$,

$$\mathrm{W}_p\left( \mathcal{R}f_\mu(\cdot, \theta), \mathcal{R}f_\nu(\cdot, \theta) \right) = \mathrm{W}_p\left( \mathcal{R}f_{g\sharp\mu}(\cdot, Q\theta), \mathcal{R}f_{g\sharp\nu}(\cdot, Q\theta) \right) \tag{37}$$

Indeed, we have

$$\mathcal{R}f_{g\sharp\mu}(t, Q\theta) = \int_{\mathbb{R}^d} f_{g\sharp\mu}(x) \cdot \delta(t - \langle x, Q\theta \rangle) \, dx \tag{38}$$

$$= \int_{\mathbb{R}^d} f_\mu(g^{-1}x) \cdot \delta(t - \langle x, Q\theta \rangle) \, dx \tag{39}$$

$$= \int_{\mathbb{R}^d} f_\mu(x) \cdot \delta(t - \langle gx, Q\theta \rangle) \, dx \tag{40}$$

$$= \int_{\mathbb{R}^d} f_\mu(x) \cdot \delta(t - \langle Qx + a, Q\theta \rangle) \, dx \tag{41}$$

$$= \int_{\mathbb{R}^d} f_\mu(x) \cdot \delta(t - \langle x + Q^{-1}a, \theta \rangle) \, dx \tag{42}$$

$$= \int_{\mathbb{R}^d} f_\mu(x) \cdot \delta(t - \langle Q^{-1}a, \theta \rangle - \langle x, \theta \rangle) \, dx \tag{43}$$

$$= \mathcal{R}f_\mu(t - \langle Q^{-1}a, \theta \rangle, \theta). \tag{44}$$

Using the closed-form of one dimensional Wasserstein distance, we have

$$\mathrm{W}_p\Big(\mathcal{R}f_{g\sharp\mu}(\cdot, Q\theta), \mathcal{R}f_{g\sharp\nu}(\cdot, Q\theta)\Big) = \mathrm{W}_p\Big(\mathcal{R}f_\mu(\cdot - \langle Q^{-1}a, \theta \rangle, \theta), \mathcal{R}f_\nu(\cdot - \langle Q^{-1}a, \theta \rangle, \theta)\Big) \tag{45}$$

$$= \mathrm{W}_p\Big(\mathcal{R}f_\mu(\cdot, \theta), \mathcal{R}f_\nu(\cdot, \theta)\Big), \tag{46}$$

where we leverage the translation invariant properties of the ground cost $L_p$-norm of $\mathrm{W}_p$ for the above second row.

So Equation (37) holds. For the rest of the proof, we show that

$$\mathrm{SW}_p(\mu, \nu) = \mathrm{SW}_p(g\sharp\mu, g\sharp\nu). \tag{47}$$

Indeed, we have

$$\mathrm{SW}_p(g\sharp\mu, g\sharp\nu) = \left( \int_{\mathbb{S}^{d-1}} \mathrm{W}_p^p(\mathcal{R}f_{g\sharp\mu}(\cdot, \theta), \mathcal{R}f_{g\sharp\nu}(\cdot, \theta)) \, d\sigma(\theta) \right)^{\frac{1}{p}} \tag{48}$$

$$= \left( \int_{\mathbb{S}^{d-1}} \mathrm{W}_p^p(\mathcal{R}f_{g\sharp\mu}(\cdot, Q\theta), \mathcal{R}f_{g\sharp\nu}(\cdot, Q\theta)) \, d\sigma(Q\theta) \right)^{\frac{1}{p}} \tag{49}$$

$$= \left( \int_{\mathbb{S}^{d-1}} \mathrm{W}_p^p(\mathcal{R}f_{g\sharp\mu}(\cdot, Q\theta), \mathcal{R}f_{g\sharp\nu}(\cdot, Q\theta)) \, d\sigma(\theta) \right)^{\frac{1}{p}} \tag{50}$$

$$= \left( \int_{\mathbb{S}^{d-1}} \mathrm{W}_p^p(\mathcal{R}f_\mu(\cdot, \theta), \mathcal{R}f_\nu(\cdot, \theta)) \, d\sigma(\theta) \right)^{\frac{1}{p}} \tag{51}$$

$$= \mathrm{SW}_p(\mu, \nu). \tag{52}$$

We finish the proof. $\square$

## B.2 $\mathcal{R}_\mathcal{L}^\alpha f$ IS INTEGRABLE

*Proof.* Let $f \in L^1(\mathbb{R}^d)$. We show that $\|\mathcal{R}_\mathcal{L}^\alpha f\|_\mathcal{L} \leqslant \|f\|_1$. Indeed,

$$\|\mathcal{R}_\mathcal{L}^\alpha f\|_\mathcal{L} = \sum_{l \in \mathcal{L}} \int_{\mathbb{R}} |\mathcal{R}_\mathcal{L}^\alpha f(t_x, l)| \, dt_x \tag{53}$$

$$= \sum_{l \in \mathcal{L}} \int_{\mathbb{R}} \left| \int_{\mathbb{R}^d} f(y) \cdot \alpha(y, \mathcal{L})_l \cdot \delta(t_x - \langle y - x_l, \theta_l \rangle) \, dy \right| dt_x \tag{54}$$

$$\leqslant \sum_{l\in\mathcal{L}} \int_{\mathbb{R}} \left( \int_{\mathbb{R}^d} |f(y)| \cdot \alpha(y,\mathcal{L})_l \cdot \delta\left(t_x - \langle y - x_l, \theta_l \rangle\right) \, dy \right) dt_x \tag{55}$$

$$= \sum_{l\in\mathcal{L}} \int_{\mathbb{R}^d} \left( \int_{\mathbb{R}} |f(y)| \cdot \alpha(y,\mathcal{L})_l \cdot \delta\left(t_x - \langle y - x_l, \theta_l \rangle\right) \, dt_x \right) dy \tag{56}$$

$$= \sum_{l\in\mathcal{L}} \int_{\mathbb{R}^d} |f(y)| \cdot \alpha(y,\mathcal{L})_l \cdot \left( \int_{\mathbb{R}} \delta\left(t_x - \langle y - x_l, \theta_l \rangle\right) \, dt_x \right) dy \tag{57}$$

$$= \sum_{l\in\mathcal{L}} \int_{\mathbb{R}^d} |f(y)| \cdot \alpha(y,\mathcal{L})_l \, dy \tag{58}$$

$$= \int_{\mathbb{R}^d} |f(y)| \cdot \sum_{l\in\mathcal{L}} \alpha(y,\mathcal{L})_l \, dy \tag{59}$$

$$= \int_{\mathbb{R}^d} |f(y)| \, dy \tag{60}$$

$$= \|f\|_1 < \infty. \tag{61}$$

So, we have $\mathcal{R}_{\mathcal{L}}^{\alpha} f \in L^1(\mathcal{L})$. It implies the operator $\mathcal{R}_{\mathcal{L}}^{\alpha} : L^1(\mathbb{R}^d) \to L^1(\mathcal{L})$ is well-defined.

Clearly, $\mathcal{R}_{\mathcal{L}}^{\alpha}$ is a linear operator. $\qquad\qquad\square$

### B.3 PROOF OF THEOREM 4.3

As we nontrivially expand the class of splitting maps from $\mathcal{C}(\mathbb{R}^d, \Delta_{k-1})$ to $\mathcal{C}(\mathbb{R}^d \times \mathbb{L}_k^d, \Delta_{k-1})$, the proof presented in (Tran et al., 2025d) is not applicable to this larger class. By introducing an invariance condition on the splitting maps, we provide a new proof for the injectivity of $\mathcal{R}^{\alpha}$ as follows.

*Proof.* Recall the notion of the original Radon Transform. Let $\mathcal{R} : L^1(\mathbb{R}^d) \to L^1(\mathbb{R} \times \mathbb{S}^{d-1})$ be the operator defined by: For $f \in L^1(\mathbb{R}^d)$, we have

$$\mathcal{R}f(t,\theta) = \int_{\mathbb{R}^d} f(y) \cdot \delta(t - \langle y, \theta \rangle) \, dy \tag{62}$$

It is well-known that the Radon Transform $\mathcal{R}$ is a linear bijection (Helgason, 2011) and its inverse $\mathcal{R}^{-1}$ is defined as

$$f(x) = \mathcal{R}^{-1}(\mathcal{R}f(t,\theta)) \tag{63}$$

$$= \int_{\mathbb{S}^{d-1}} (\mathcal{R}f(\langle x,\theta \rangle, \theta) * \eta) (\langle x,\theta \rangle) \, d\theta, \tag{64}$$

where the convolution kernel $\eta$ satisfies that its Fourier transform $\hat{\eta}(\omega) = |\omega|$.

Back to the problem. Recall that $\mathbb{L}_k^d$ is the collection of all systems of $k$ lines in $\mathbb{R}^d \times \mathbb{S}^{d-1}$,

$$\mathbb{L}_k^d = \left\{ \mathcal{L} = \{(x_j, \theta_j)\}_{j=1}^k : x_j \in \mathbb{R}^d, \theta_j \in \mathbb{S}^{d-1} \right\} = \left( \mathbb{R}^d \times \mathbb{S}^{d-1} \right)^k. \tag{65}$$

For an index $i$ such that $1 \leqslant i \leqslant k$ and a direction $\theta \in \mathbb{S}^{d-1}$, define

$$\mathbb{L}_k^d(i,\theta) := \left\{ \mathcal{L} = \{(x_j, \theta_j)\}_{j=1}^k \in \left( \mathbb{R}^d \times \mathbb{S}^{d-1} \right)^k : \theta_i = \theta \right\}. \tag{66}$$

In words, $\mathbb{L}_k^d(i,\theta)$ is a subcollection of $\mathbb{L}_k^d$ consists of all systems of $k$ lines with the direction of the $i^{\text{th}}$ line is equal to $\theta$. It is clear that $\mathbb{L}_k^d$ is the disjoint union of all $\mathbb{L}_k^d(i,\theta)$ for $\theta \in \mathbb{S}^{d-1}$,

$$\mathbb{L}_k^d = \bigsqcup_{\theta \in \mathbb{S}^{d-1}} \mathbb{L}_k^d(i,\theta). \tag{67}$$

We have some observations on these subcollections.

**Result 1.** Each $g = (Q, a) \in E(d)$, define a bijection between $\mathbb{L}_k^d(i, \theta)$ and $\mathbb{L}_k^d(i, Q \cdot \theta)$. More precisely, for the map $\phi_g$, defined by

$$\phi_g : \qquad \mathbb{L}_k^d(i, \theta) \qquad \longrightarrow \qquad \mathbb{L}_k^d(i, Q \cdot \theta) \tag{68}$$

$$\mathcal{L} = \{(x_j, \theta_j)\}_{j=1}^k \longmapsto g\mathcal{L} = \{(Q \cdot x_j + a, Q \cdot \theta_j)\}_{j=1}^k, \tag{69}$$

it is well-defined and is a bijection. This can be directly verified by definitions.

**Result 2.** For all $1 \leqslant i \leqslant k$, $y, y' \in \mathbb{R}^d$ and $\theta, \theta' \in \mathbb{S}^{d-1}$, we have

$$\int_{\mathbb{L}_k^d(i,\theta)} \alpha(y, \mathcal{L})_i \, d\mathcal{L} = \int_{\mathbb{L}_k^d(i,\theta')} \alpha(y', \mathcal{L})_i \, d\mathcal{L}. \tag{70}$$

Note that, the integrations are taken over $\mathbb{L}_k^d(i, \theta)$ and $\mathbb{L}_k^d(i, \theta')$ with measures induced from the measure of $\mathbb{L}_k^d$. We show that Equation (70) holds. Note that, for $\theta, \theta' \in \mathbb{S}^{d-1}$, there exists an orthogonal transformation $Q \in O(d)$ such that $Q \cdot \theta = \theta'$. Let $a = y' - Q \cdot y$, and $g = (Q, a) \in E(d)$. By this definition, we have

$$gy = Q \cdot y + a = Q \cdot y + y' - Q \cdot y = y'. \tag{71}$$

From **Result 1**, we have a corresponding bijection $\phi_g$ from $\mathbb{L}_k^d(i, \theta)$ to $\mathbb{L}_k^d(i, \theta')$. We have

$$\int_{\mathbb{L}_k^d(i,\theta')} \alpha(y', \mathcal{L})_i \, d\mathcal{L} = \int_{\mathbb{L}_k^d(i,\theta)} \alpha(y', g\mathcal{L})_i \, d(g\mathcal{L}) \qquad \text{(change of variables)} \tag{72}$$

$$= \int_{\mathbb{L}_k^d(i,\theta)} \alpha(gy, g\mathcal{L})_i \, d(g\mathcal{L}) \qquad \text{(since } y' = gy\text{)} \tag{73}$$

$$= \int_{\mathbb{L}_k^d(i,\theta)} \alpha(y, \mathcal{L})_i \, d(g\mathcal{L}) \qquad \text{(since } \alpha \text{ is } E(d)\text{-invariant)} \tag{74}$$

$$= \int_{\mathbb{L}_k^d(i,\theta)} \alpha(y, \mathcal{L})_i \, d\mathcal{L} \qquad \text{(since } |\det(Q)| = 1\text{)} \tag{75}$$

We finish the proof for **Result 2**.

**Result 3.** From **Result 2**, for all $1 \leqslant i \leqslant k$, we can define a constant $c_i$ such that

$$c_i := \int_{\mathbb{L}_k^d(i,\theta)} \alpha(y, \mathcal{L})_i \, d\mathcal{L}. \tag{76}$$

for all $y \in \mathbb{R}^d$ and $\theta \in \mathbb{S}^{d-1}$. Then

$$c_1 + c_2 + \ldots + c_k = 1. \tag{77}$$

In particular, there exists $1 \leqslant i \leqslant k$ such that $c_i$ is non-zero. To show this, first, recall that $\mathbb{L}_k^d$ is the disjoint union of all $\mathbb{L}_k^d(i, \theta)$ for $\theta \in \mathbb{S}^{d-1}$,

$$\mathbb{L}_k^d = \bigsqcup_{\theta \in \mathbb{S}^{d-1}} \mathbb{L}_k^d(i, \theta), \tag{78}$$

so we have

$$\int_{\mathbb{L}_k^d} \alpha(y, \mathcal{L})_i \, d\mathcal{L} = \int_{\mathbb{S}^{d-1}} \left( \int_{\mathbb{L}_k^d(i,\theta)} \alpha(y, \mathcal{L})_i \, d\mathcal{L} \right) d\theta \tag{79}$$

$$= \int_{\mathbb{S}^{d-1}} c_i \, d\theta \tag{80}$$

$$= c_i. \tag{81}$$

Then

$$c_1 + c_2 + \ldots c_k = \sum_{j=1}^k \int_{\mathbb{L}_k^d} \alpha(y, \mathcal{L})_j \, d\mathcal{L} \tag{82}$$

$$= \int_{\mathbb{L}_k^d} \left( \sum_{j=1}^{k} \alpha(y, \mathcal{L})_j \right) d\mathcal{L} \tag{83}$$

$$= \int_{\mathbb{L}_k^d} 1 \, d\mathcal{L} \tag{84}$$

$$= 1. \tag{85}$$

We finish the proof for **Result 3**.

Consider a splitting map $\alpha$ in $\mathcal{C}(\mathbb{R}^d \times \mathbb{L}_k^d, \Delta_{k-1})$ that is E($d$)-invariant. By **Result 3**, let $1 \leqslant i \leqslant k$ is the index that

$$c_i = \int_{\mathbb{L}_k^d(i,\theta)} \alpha(y, \mathcal{L})_i \, d\mathcal{L} \neq 0. \tag{86}$$

Now, for a system of lines $\mathcal{L}$ in $\mathbb{L}_k^d$, we denote the $i^{\text{th}}$ line by $l_{\mathcal{L}:i}$ and its source by $x_{\mathcal{L}:i}$. For a function $f \in L^1(\mathbb{R}^d)$, define a function $g \in L^1(\mathbb{R} \times \mathbb{S}^{d-1})$ as follows

$$g : \quad \mathbb{R} \times \mathbb{S}^{d-1} \longrightarrow \mathbb{R} \tag{87}$$

$$(t, \theta) \quad \longmapsto \int_{\mathbb{L}_k^d(i,\theta)} \mathcal{R}_{\mathcal{L}}^\alpha f\big(t - \langle x_{\mathcal{L}:i}, \theta \rangle, l_{\mathcal{L}:i}\big) \, d\mathcal{L} \tag{88}$$

From the definition of $\mathcal{R}_{\mathcal{L}}^\alpha f$,

$$\mathcal{R}_{\mathcal{L}}^\alpha f : \quad \bar{\mathcal{L}} \longrightarrow \mathbb{R} \tag{89}$$

$$(x, l) \longmapsto \int_{\mathbb{R}^d} f(y) \cdot \alpha(y, \mathcal{L})_l \cdot \delta\left(t_x - \langle y - x_l, \theta_l \rangle\right) \, dy, \tag{90}$$

we have

$$g(t, \theta) = \int_{\mathbb{L}_k^d(i,\theta)} \mathcal{R}_{\mathcal{L}}^\alpha f\big(t - \langle x_{\mathcal{L}:i}, \theta \rangle, l_{\mathcal{L}:i}\big) \, d\mathcal{L} \tag{91}$$

$$= \int_{\mathbb{L}_k^d(i,\theta)} \left( \int_{\mathbb{R}^d} f(y) \cdot \alpha(y, \mathcal{L})_i \cdot \delta\left(t - \langle x_{\mathcal{L}:i}, \theta \rangle - \langle y - x_{\mathcal{L}:i}, \theta \rangle\right) \, dy \right) d\mathcal{L} \tag{92}$$

$$= \int_{\mathbb{L}_k^d(i,\theta)} \left( \int_{\mathbb{R}^d} f(y) \cdot \alpha(y, \mathcal{L})_i \cdot \delta\left(t - \langle x_{\mathcal{L}:i} + y - x_{\mathcal{L}:i}, \theta \rangle\right) \, dy \right) d\mathcal{L} \tag{93}$$

$$= \int_{\mathbb{L}_k^d(i,\theta)} \left( \int_{\mathbb{R}^d} f(y) \cdot \alpha(y, \mathcal{L})_i \cdot \delta\left(t - \langle y, \theta \rangle\right) \, dy \right) d\mathcal{L} \tag{94}$$

$$= \int_{\mathbb{R}^d} \left( \int_{\mathbb{L}_k^d(i,\theta)} f(y) \cdot \alpha(y, \mathcal{L})_i \cdot \delta\left(t - \langle y, \theta \rangle\right) \, d\mathcal{L} \right) dy \tag{95}$$

$$= \int_{\mathbb{R}^d} f(y) \cdot \delta\left(t - \langle y, \theta \rangle\right) \cdot \left( \int_{\mathbb{L}_k^d(i,\theta)} \alpha(y, \mathcal{L})_i \, d\mathcal{L} \right) dy \tag{96}$$

$$= c_i \cdot \int_{\mathbb{R}^d} f(y) \cdot \delta\left(t - \langle y, \theta \rangle\right) \, dy \tag{97}$$

$$= c_i \cdot \mathcal{R}f(t, \theta). \tag{98}$$

Let $f \in \text{Ker } \mathcal{R}^\alpha$, which means $\mathcal{R}_{\mathcal{L}}^\alpha f = 0$ for all $\mathcal{L} \in \mathbb{L}_k^d$. So $g = 0 \in L^1(\mathbb{R} \times \mathbb{S}^{d-1})$, and since $c_i \neq 0$, it implies $\mathcal{R}f = 0 \in L^1(\mathbb{R} \times \mathbb{S}^{d-1})$. Additionally, recall that the Radon Transform $\mathcal{R}$ is a bijection, we conclude that $f = 0 \in L^1(\mathbb{R}^d)$.

Hence, $\mathcal{R}^\alpha$ is injective. The proof is completed. $\qquad\square$

*Remark* 9. A formal proof would require Haar measure theory for compact groups, but we simplify this for brevity and relevance to the scope of the paper.

*Remark* 10. The injectivity still hold if we restrict $\mathbb{L}_k^d$ to a non-empty subset of $\mathbb{L}_k^d$ that is closed under action of $\mathrm{E}(d)$. In concrete, let $A$ be a non-empty subset of $\mathbb{L}_k^d$ satisfies that $g\mathcal{L} \in A$ for all $g \in \mathrm{E}(d)$ and $\mathcal{L} \in A$. Let $f \in L^1(\mathbb{R}^d)$ such that $\mathcal{R}_{\mathcal{L}}^\alpha f = 0$ for all $\mathcal{L} \in A$. Using the same argument, we can demonstrate that $f = 0$. In particular, for $\mathbb{T}$ and $\mathbb{T}^\perp$ are introduced in Subsection 5.2, since both $\mathbb{T}$ and $\mathbb{T}^\perp$ are closed under action of $\mathrm{E}(d)$, we see that a function $f \in L^1(\mathbb{R}^d)$ is equal to 0, if $\mathcal{R}_{\mathcal{L}}^\alpha f = 0$ for all $\mathcal{L} \in \mathbb{T}$, or for all $\mathcal{L} \in \mathbb{T}^\perp$.

### B.4    PROOF OF THEOREM 5.2

*Proof.* We show that

$$\text{Db-TSW}(\mu, \nu) = \int_{\mathbb{T}} W_{d_{\mathcal{L}},1}(\mathcal{R}_{\mathcal{L}}^\alpha \mu, \mathcal{R}_{\mathcal{L}}^\alpha \nu) \, d\sigma(\mathcal{L})., \tag{99}$$

is a metric on $\mathcal{P}(\mathbb{R}^d)$.

**Positive definiteness.**    For $\mu, \nu \in \mathcal{P}(\mathbb{R}^n)$, it is clear that $\text{Db-TSW}(\mu, \mu) = 0$ and $\text{Db-TSW}(\mu, \nu) \geqslant 0$. If $\text{Db-TSW}(\mu, \nu) = 0$, then $W_{d_{\mathcal{L}},1}(\mathcal{R}_{\mathcal{L}}^\alpha \mu, \mathcal{R}_{\mathcal{L}}^\alpha \nu) = 0$ for all $\mathcal{L} \in \mathbb{T}$. Since $W_{d_{\mathcal{L}},1}$ is a metric on $\mathcal{P}(\mathcal{L})$, we have $\mathcal{R}_{\mathcal{L}}^\alpha \mu = \mathcal{R}_{\mathcal{L}}^\alpha \nu$ for all $\mathcal{L} \in \mathbb{T}$. By the remark at the end of Appendix B.3, it implies that $\mu = \nu$.

**Symmetry.**    For $\mu, \nu \in \mathcal{P}(\mathbb{R}^n)$, we have:

$$\text{Db-TSW}(\mu, \nu) = \int_{\mathbb{T}} W_{d_{\mathcal{L}},1}(\mathcal{R}_{\mathcal{L}}^\alpha \mu, \mathcal{R}_{\mathcal{L}}^\alpha \nu) \, d\sigma(\mathcal{L}) \tag{100}$$

$$= \int_{\mathbb{T}} W_{d_{\mathcal{L}},1}(\mathcal{R}_{\mathcal{L}}^\alpha \nu, \mathcal{R}_{\mathcal{L}}^\alpha \mu) \, d\sigma(\mathcal{L}) = \text{Db-TSW}(\nu, \mu). \tag{101}$$

So $\text{Db-TSW}(\mu, \nu) = \text{TSW-SL}(\nu, \mu)$.

**Triangle inequality.**    For $\mu_1, \mu_2, \mu_3 \in \mathcal{P}(\mathbb{R}^n)$, we have:

$$\text{Db-TSW}(\mu_1, \mu_2) + \text{Db-TSW}(\mu_2, \mu_3) \tag{102}$$

$$= \int_{\mathbb{T}} W_{d_{\mathcal{L}},1}(\mathcal{R}_{\mathcal{L}}^\alpha \mu_1, \mathcal{R}_{\mathcal{L}}^\alpha \mu_2) \, d\sigma(\mathcal{L}) + \int_{\mathbb{T}} W_{d_{\mathcal{L}},1}(\mathcal{R}_{\mathcal{L}}^\alpha \mu_2, \mathcal{R}_{\mathcal{L}}^\alpha \mu_3) \, d\sigma(\mathcal{L}) \tag{103}$$

$$= \int_{\mathbb{T}} \left( W_{d_{\mathcal{L}},1}(\mathcal{R}_{\mathcal{L}}^\alpha \mu_1, \mathcal{R}_{\mathcal{L}}^\alpha \mu_2) \, d\sigma(\mathcal{L}) + W_{d_{\mathcal{L}},1}(\mathcal{R}_{\mathcal{L}}^\alpha \mu_2, \mathcal{R}_{\mathcal{L}}^\alpha \mu_3) \right) d\sigma(\mathcal{L}) \tag{104}$$

$$\geqslant \int_{\mathbb{T}} W_{d_{\mathcal{L}},1}(\mathcal{R}_{\mathcal{L}}^\alpha \mu_1, \mathcal{R}_{\mathcal{L}}^\alpha \mu_3) \, d\sigma(\mathcal{L}) \tag{105}$$

$$= \text{Db-TSW}(\mu_1, \mu_3). \tag{106}$$

So the triangle inequality holds for Db-TSW.

In conclusion, Db-TSW is a metric on the space $\mathcal{P}(\mathbb{R}^d)$.

**Db-TSW is $\mathrm{E}(d)$-invariant.**    We need to show that Db-TSW is $\mathrm{E}(d)$-invariant, which means for all $g \in \mathrm{E}(d)$ such that

$$\text{Db-TSW}(\mu, \nu) = \text{Db-TSW}(g\sharp\mu, g\sharp\nu), \tag{107}$$

where $g\sharp\mu, g\sharp\nu$ as the pushforward of $\mu, \nu$ via Euclidean transformation $g \colon \mathbb{R}^d \to \mathbb{R}^d$, respectively. For a tree system $\mathcal{L} \in \mathbb{T}$ such that $\mathcal{L} = \left\{ l_i = (x_i, \theta_i) \right\}_{i=1}^k$, we have

$$g\mathcal{L} = \left\{ gl_i = (Q \cdot x_i + a, Q \cdot \theta_i) \right\}_{i=1}^k. \tag{108}$$

For $g = (Q, a)$, note that $|\det(Q)| = 1$, we have

$$\mathcal{R}_{g\mathcal{L}}^\alpha(g\sharp\mu)(gx, gl) = \int_{\mathbb{R}^d} (g\sharp\mu)(y) \cdot \alpha(y, g\mathcal{L})_l \cdot \delta\left(t_{gx} - \langle y - x_{gl}, \theta_{gl} \rangle\right) \, dy \tag{109}$$

$$= \int_{\mathbb{R}^d} \mu(g^{-1}y) \cdot \alpha(y, g\mathcal{L})_l \cdot \delta\left(t_x - \langle y - x_{gl}, \theta_{gl} \rangle\right) \, dy \tag{110}$$

$$= \int_{\mathbb{R}^d} \mu(g^{-1}gy) \cdot \alpha(gy, g\mathcal{L})_l \cdot \delta\left(t_x - \langle gy - x_{gl}, \theta_{gl} \rangle\right) \, d(gy) \tag{111}$$

$$= \int_{\mathbb{R}^d} \mu(y) \cdot \alpha(y, \mathcal{L})_l \cdot \delta\left(t_x - \langle gy - x_{gl}, \theta_{gl} \rangle\right) \, dy \tag{112}$$

$$= \int_{\mathbb{R}^d} \mu(y) \cdot \alpha(y, \mathcal{L})_l \cdot \delta\left(t_x - \langle Q \cdot y + a - Q \cdot x_l - a, Q \cdot \theta_l \rangle\right) \, dy \tag{113}$$

$$= \int_{\mathbb{R}^d} \mu(y) \cdot \alpha(y, \mathcal{L})_l \cdot \delta\left(t_x - \langle Q \cdot y - Q \cdot x_l, Q \cdot \theta_l \rangle\right) \, dy \tag{114}$$

$$= \int_{\mathbb{R}^d} \mu(y) \cdot \alpha(y, \mathcal{L})_l \cdot \delta\left(t_x - \langle y - x_l, \theta_l \rangle\right) \, dy \tag{115}$$

$$= \mathcal{R}^\alpha_{\mathcal{L}} \mu(x, l) \tag{116}$$

Similarly, we have

$$\mathcal{R}^\alpha_{g\mathcal{L}}(g\sharp\nu)(gx, gl) = \mathcal{R}^\alpha_{\mathcal{L}}\nu(x, l). \tag{117}$$

Moreover, $g$ induces an isometric transformation $\mathcal{L} \to g\mathcal{L}$, so

$$\mathrm{W}_{d_\mathcal{L}, 1}(\mathcal{R}^\alpha_{\mathcal{L}}\mu, \mathcal{R}^\alpha_{\mathcal{L}}\nu) = \mathrm{W}_{d_{g\mathcal{L}}, 1}(\mathcal{R}^\alpha_{g\mathcal{L}}g\sharp\mu, \mathcal{R}^\alpha_{g\mathcal{L}}g\sharp\nu). \tag{118}$$

Finally, we have

$$\mathrm{Db\text{-}TSW}(g\sharp\mu, g\sharp\nu) = \int_{\mathbb{L}^d_k} \mathrm{W}_{d_\mathcal{L}, 1}(\mathcal{R}^\alpha_{\mathcal{L}}g\sharp\mu, \mathcal{R}^\alpha_{\mathcal{L}}g\sharp\nu) \, d\sigma(\mathcal{L}) \tag{119}$$

$$= \int_{\mathbb{T}} \mathrm{W}_{d_{g\mathcal{L}}, 1}(\mathcal{R}^\alpha_{g\mathcal{L}}g\sharp\mu, \mathcal{R}^\alpha_{g\mathcal{L}}g\sharp\nu) \, d\sigma(g\mathcal{L}) \tag{120}$$

$$= \int_{\mathbb{T}} \mathrm{W}_{d_\mathcal{L}, 1}(\mathcal{R}^\alpha_{\mathcal{L}}\mu, \mathcal{R}^\alpha_{\mathcal{L}}\nu) \, d\sigma(\mathcal{L}) \tag{121}$$

$$= \mathrm{Db\text{-}TSW}(\mu, \nu) \tag{122}$$

We conclude that Db-TSW is $\mathrm{E}(d)$-invariant. $\qquad\square$

*Remark* 11. We will omit the "almost-surely conditions" in the proofs, as they are simple to verify and their inclusion would only introduce unnecessary complexity.

## C EXPERIMENTAL DETAILS

### C.1 ALGORITHM OF DB-TSW

---

**Algorithm 1** Distance-based Tree-Sliced Wasserstein distance.

---

**Input:** Probability measures $\mu$ and $\nu$ in $\mathcal{P}(\mathbb{R}^d)$, number of tree systems $L$, number of lines in tree system $k$, space of tree systems $\mathbb{T}$ or $\mathbb{T}^\perp$, splitting maps $\alpha$ with tuning parameter $\delta \in \mathbb{R}$.
**for** $i = 1$ to $L$ **do**
    Sampling $x \in \mathbb{R}^d$ and $\theta_1, \ldots, \theta_k \overset{i.i.d}{\sim} \mathcal{U}(\mathbb{S}^{d-1})$.
    **if** space of tree system is $\mathcal{T}^\perp$ **then**
        Orthonormalize $\theta_1, \ldots, \theta_k$.
    **end if**
    Contruct tree system $\mathcal{L}_i = \{(x, \theta_1), \ldots, (x, \theta_k)\}$.
    Projecting $\mu$ and $\nu$ onto $\mathcal{L}_i$ to get $\mathcal{R}^\alpha_{\mathcal{L}_i}\mu$ and $\mathcal{R}^\alpha_{\mathcal{L}_i}\nu$.
    Compute $\widehat{\mathrm{Db\text{-}TSW}}(\mu, \nu) = (1/L) \cdot \mathrm{W}_{d_{\mathcal{L}_i}, 1}(\mathcal{R}^\alpha_{\mathcal{L}_i}\mu, \mathcal{R}^\alpha_{\mathcal{L}_i}\nu)$.
**end for**
**Return:** $\widehat{\mathrm{Db\text{-}TSW}}(\mu, \nu)$.

---

## C.2   Denoising Diffusion Models

**Diffusion Models.**   Diffusion models (Sohl-Dickstein et al., 2015; Ho et al., 2020) have gained significant attention for their ability to generate high-quality data. In this experiment, we explore how these models work and the improvements introduced by our method. The diffusion process starts with a sample from distribution $q(x_0)$ and gradually adding Gaussian noise to data $x_0$ over $T$ steps, described by $q(x_{1:T}|x_0) = \prod_{t=1}^{T} q(x_t|x_{t-1})$, where $q(x_t|x_{t-1}) = \mathcal{N}(x_t; \sqrt{1-\beta_t}x_{t-1}, \beta_t I)$ with a predefined variance schedule $\beta_t$.

The denoising diffusion model aim to learns the reverse diffusion process to effectively reconstruct the original data from noisy observations. The training process of the denoising diffusion model aim to estimate the parameters $\theta$ of the reverse process, which is defined by $p_\theta(x_{0:T}) = p(x_T) \prod_{t=1}^{T} p_\theta(x_{t-1}|x_t)$, where $p_\theta(x_{t-1}|x_t) = \mathcal{N}(x_{t-1}; \mu_\theta(x_t, t), \sigma_t^2 I)$. The standard training approach uses maximum likelihood by optimizing the evidence lower bound (ELBO), $\tilde{\mathcal{L}} \leqslant p_\theta(x_0)$, which aims to minimize the Kullback-Leibler divergence between the true posterior and the model's approximation of the reverse diffusion process across all time steps:

$$\tilde{\mathcal{L}} = -\sum_{t=1}^{T} \mathbb{E}_{q(x_t)} \left[ \text{KL}(q(x_{t-1}|x_t)||p_\theta(x_{t-1}|x_t)) \right] + C,$$

where KL refers to the Kullback-Leibler divergence, and $C$ represents a constant term.

**Denoising Diffusion GANs.**   Original diffusion models, while producing high-quality and diverse samples, are limited by their slow sampling process, which hinders their applications in real-world scenarios. Denoising diffusion GANs (Xiao et al., 2022) address this limitation by modeling each denoising step with a multimodal conditional GAN, enabling larger denoising steps and thus significantly reducing the total number of steps required to $4$, resulting in sampling speeds up to $2000$ times faster than traditional diffusion models while maintaining competitive sample quality and diversity. Denoising diffusion GANs introduce an implicit denoising model:

$$p_\theta(x_{t-1}|x_t) = \int p_\theta(x_{t-1}|x_t, \epsilon) G_\theta(x_t, \epsilon) d\epsilon, \quad \epsilon \sim \mathcal{N}(0, I).$$

Xiao et al. (2022) employ adversarial training to optimize the model parameters $\theta$. Its loss is defined by:

$$\min_\phi \sum_{t=1}^{T} \mathbb{E}_{q(x_t)} [D_{adv}(q(x_{t-1}|x_t)||p_\phi(x_{t-1}|x_t))],$$

where $D_{adv}$ is the adversarial loss. Nguyen et al. (2024b) replace the adversarial loss by the augmented generalized Mini-batch Energy distance. For two distributions $\mu$ and $\nu$, with a mini-batch size $n \geqslant 1$, the augmented generalized mini-batch Energy distance (AGME) using a Sliced Wasserstein (SW) kernel is expressed as:

$$\text{AGME}_b^2(\mu, \nu; g) = \text{GME}_b^2(\tilde{\mu}, \tilde{\nu}),$$

where $\tilde{\mu} = f_\# \mu$ and $\tilde{\nu} = f_\# \nu$ with $f(x) = (x, g(x))$ for a nonlinear function $g : \mathbb{R}^d \to \mathbb{R}$. GME is the generalized Mini-batch Energy distance (Salimans et al., 2018), defined as

$$\text{GME}_b^2(\mu, \nu) = 2\mathbb{E}[D(P_X, P_Y)] - \mathbb{E}[D(P_X, P_X')] - \mathbb{E}[D(P_Y, P_Y')],$$

where $X, X' \overset{i.i.d.}{\sim} \mu^{\otimes m}$ and $Y, Y' \overset{i.i.d.}{\sim} \nu^{\otimes m}$, with $P_X = \frac{1}{m} \sum_{i=1}^{m} \delta_{x_i}$, $X = (x_1, \dots, x_m)$. and $D$ are any valid distance. Here we replace $D$ by TSW variants (including our proposed Db-TSW) and SW variants.

**Setting**   We use the same architecture and hyperparameters as in Nguyen et al. (2024b) and train our models over 1800 epochs. For all our methods, including Db-TSW-DD$^\perp$ and Db-TSW-DD, we use $L = 2500, k = 4, \delta = 10$. For vanilla SW and SW variants, we follow Nguyen et al. (2024b) to use $L = 10000$.

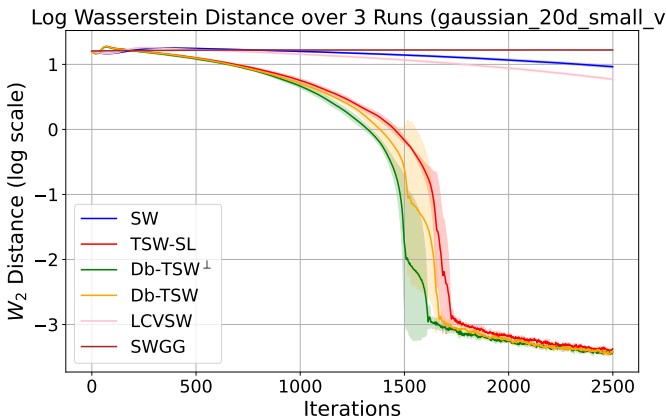

Figure 4: Logarithm of Wasserstein Distance over 3 runs on Gaussian 20d dataset.

Table 3: Average Wasserstein distance between source and target distributions of 10 runs on 25 Gaussians dataset. All methods use 100 projecting directions.

| Methods | Iteration | | | | | Time/Iter($s$) |
| --- | --- | --- | --- | --- | --- | --- |
| | 500 | 1000 | 1500 | 2000 | 2500 | |
| SW | **1.61e-1** | 9.52e-2 | 3.44e-2 | 2.56e-2 | 2.20e-2 | 0.002 |
| MaxSW | 5.09e-1 | 2.36e-1 | 1.33e-1 | 9.70e-2 | 8.48e-2 | 0.144 |
| SWGG | 3.10e-1 | 1.17e-1 | 3.38e-2 | 3.58e-3 | 2.54e-4 | 0.002 |
| LCVSW | 3.38e-1 | 6.64e-2 | 3.06e-2 | 3.06e-2 | 3.02e-2 | 0.001 |
| TSW-SL | 3.49e-1 | 9.06e-2 | 2.96e-2 | 1.20e-2 | 3.03e-7 | 0.002 |
| Db-TSW-DD | 3.84e-1 | 1.13e-1 | **2.48e-2** | 2.96e-3 | 1.00e-7 | 0.002 |
| Db-TSW-DD$^\perp$ | 3.82e-1 | **1.11e-1** | 2.73e-2 | **1.45e-3** | **9.97e-8** | 0.003 |

## C.3  GRADIENT FLOW

**Additional Results on Multi-modal Synthetic Dataset.**   Table 3 showcases the performance of our methods on the Gradient Flow task using the 25 Gaussians (multi-modal distribution) dataset. Our approach consistently achieves the smallest Wasserstein distance, demonstrating robust performance across various dataset types, including non-linear, multi-modal, and high-dimensional cases. These results indicate that our methods outperform baseline models, further highlighting their effectiveness across different challenging scenarios.

**Hyperparameters.**   For Db-TSW$^\perp$ and Db-TSW, we use $L = 25, k = 4, \delta = 10$ for all our experiments. For the baselines of SW and SW-variants, we use $L = 100$. The number of supports generated for each distribution in all datasets is 100.

We follow Mahey et al. (2023) to set the global learning rate for all baselines to be 5e-3 for all datasets. For our methods, we use the global learning rate of 5e-3 for 25 Gaussians and Swiss Roll datasets and 5e-2 for Gaussian 20d.

**Gaussian 20d detail results.** We show the detail result of Db-TSW and Db-TSW$^\perp$ in Table 4 and visualize the result in Figure 4. Figure 4 reveals distinct performance patterns among different methods, with TSW variants showing faster convergence rate. Notably, Db-TSW$^\perp$ achieves the fastest convergence, followed by Db-TSW and TSW-SL, all exhibiting a sharp performance improvement around iteration 1500. While these methods show some fluctuation mid-process as indicated by the shaded variance regions, they maintain remarkable stability in the final iterations (iteration 2000–2500). These results are further supported by the quantitative mean and standard deviation values provided in Table 4.

Table 4: Wasserstein distance between source and target distributions of 3 runs on Gaussian 20d dataset. All methods use 100 projecting directions.

| Methods | Iteration | | | | |
|---|---|---|---|---|---|
| | 500 | 1000 | 1500 | 2000 | 2500 |
| SW | $17.57 \pm 2.4$e-1 | $15.86 \pm 3.1$e-1 | $13.92 \pm 3.7$e-1 | $11.70 \pm 4.2$e-1 | $9.22 \pm 3.8$e-1 |
| SWGG | $16.53 \pm 1.2$e-1 | $16.64 \pm 1.4$e-1 | $16.65 \pm 1.7$e-1 | $16.63 \pm 1.6$e-1 | $16.64 \pm 1.5$e-1 |
| LCVSW | $16.86 \pm 4$e-1 | $14.36 \pm 3.4$e-1 | $11.68 \pm 3.3$e-1 | $8.80 \pm 4$e-1 | $5.91 \pm 2.3$e-1 |
| TSW-SL | $12.61 \pm 3$e-1 | $5.68 \pm 3.8$e-1 | $6.90$e-1 $\pm 8$e-2 | $6.83$e-4 $\pm 2.6$e-5 | $4.22$e-4 $\pm 1.2$e-5 |
| Db-TSW | $\underline{12.46} \pm 4.6$e-1 | $\underline{5.12} \pm 3.9$e-1 | $\underline{4.8\text{e-1}} \pm 3.3$e-1 | $\underline{6.08\text{e-4}} \pm 6.7$e-5 | $\underline{3.84\text{e-4}} \pm 2.2$e-5 |
| Db-TSW$^{\perp}$ | $\mathbf{12.15} \pm 3.5$e-1 | $\mathbf{4.67} \pm 3.6$e-1 | $\mathbf{1.22\text{e-1}} \pm 1.6$e-1 | $\mathbf{5.74\text{e-4}} \pm 2$e-5 | $\mathbf{3.78\text{e-4}} \pm 1.4$e-5 |

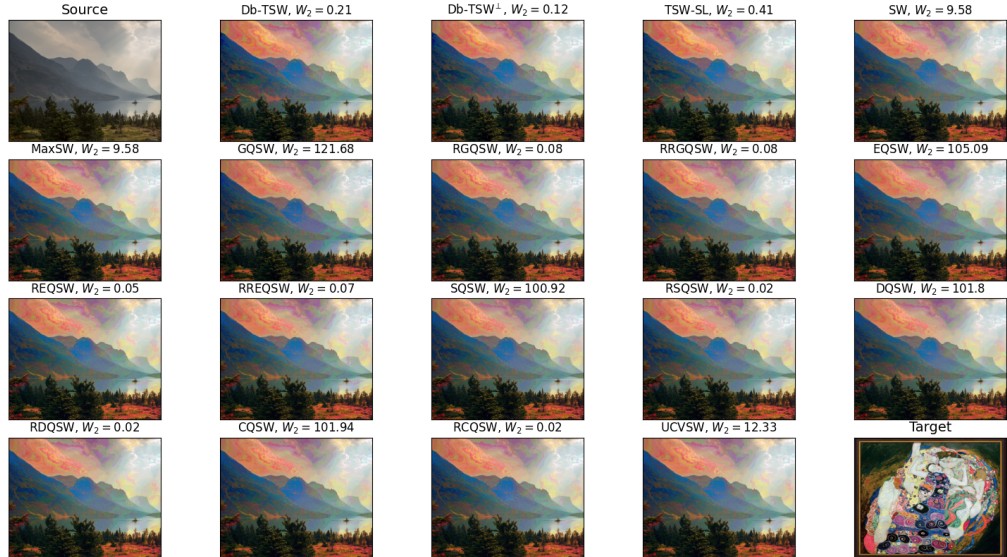

Figure 5: Comparison of color transferred image.

## C.4 COLOR TRANSFER

**Color transfer.** Given a source image and a target image, we represent their respective color palettes as matrices $X$ and $Y$, where each matrix is characterized by dimensions $n \times 3$ (with $n$ denoting the number of pixels). The source and target images employed in this analysis are derived from the work of Nguyen et al. (2024a). This curve is initialized at $Z(0) = X$ and is terminated at $Z(T) = Y$. Subsequently, we reduce the number of distinct colors in both images to 1000 utilizing K-means clustering. We then trace the trajectory between the empirical distributions of colors in the source image ($P_X$) and the target image ($P_Y$) through an approximate Euler integration scheme. Given that the RGB color values are constrained within the range $\{0, \dots, 255\}$, a rounding procedure is implemented at each Euler step during the final iterations.

**Hyperparameters.** The total number of iterations for all methodologies is standardized at 2000. A step size of 1 is utilized for all baseline methods, while a step size of 17 is adopted for TSW-SL, Db-TSW-SL$^{\perp}$, and Db-TSW-SL. For the sliced Wasserstein variants, we set $L = 100$, whereas in TSW-SL and our proposed methods, we utilize $L = 33$ and $k = 3$ to ensure a consistent and fair comparison.

**Qualitative results.** Figure 6 shows the qualitative results of our methods, including Db-TSW-DD and Db-TSW-DD$^{\perp}$.

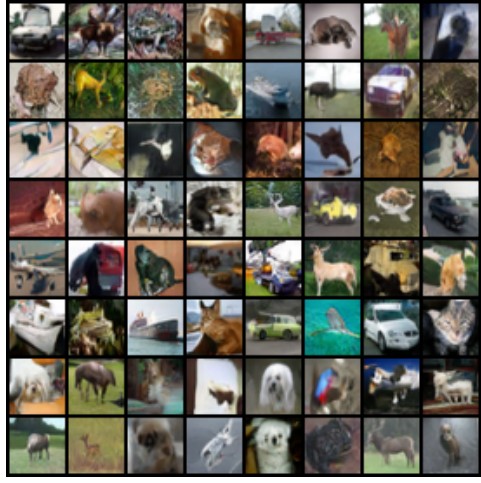 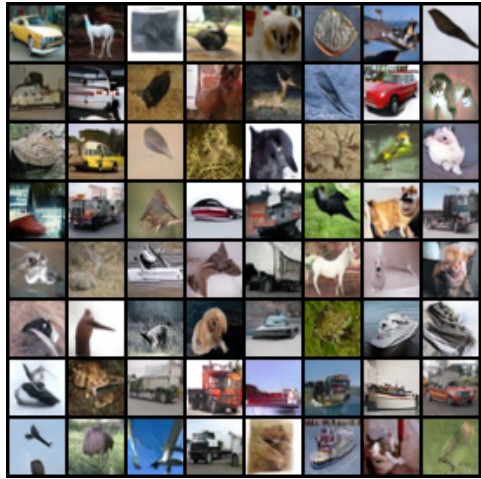

(a) Generated images of Db-TSW-DD

(b) Generated images of Db-TSW-DD$^\perp$

Figure 6: Comparison of generated images from Db-TSW-DD and Db-TSW-DD$^\perp$ methods.

## C.5 COMPUTATIONAL INFRASTRUCTURE

The experiments of gradient flow and color transfer were carried out on a single NVIDIA A100 GPU. Each experiments for gradient flow take roughly $0.5$ hours. The runtime for color transfer experiments is $5$ minutes.

In contrast, the denoising diffusion experiments were executed in parallel on two NVIDIA A100 GPUs, with each run lasting around $81.5$ hours.

## C.6 COMPUTATION AND MEMORY COMPLEXITY OF DB-TSW

Since GPUs are now standard in machine learning workflows due to their parallel processing capabilities, and Db-TSW is primarily intended use for deep learning tasks like generative modeling and optimal transport problems which are typically GPU-accelerated, we provide both complexity analysis and empirical runtime/memory of Db-TSW with GPU settings.

**Computation and memory complexity of Db-TSW.** Assume $n \geqslant m$, the computational complexity and memory complexity of Db-TSW are $O(Lknd + Lkn \log n)$ and $O(Lkn + Lkd + nd)$, respectively. In details, the Db-TSW algorithm will have there most costly operations as described in Table 5:

Table 5: Complexity Analysis of Db-TSW

| Operation | Description | Computation | Memory |
|---|---|---|---|
| Projection | Matrix multiplication of points and lines | $O(Lknd)$ | $O(Lkd + nd)$ |
| Distance-based weight splitting | Distance calculation and softmax | $O(Lknd)$ | $O(Lkn + Lkd + nd)$ |
| Sorting | Sorting projected coordinates | $O(Lkn \log n)$ | $O(Lkn)$ |
| **Total** | | $O(Lknd + Lkn \log n)$ | $O(Lkn + Lkd + nd)$ |

**The kernel fusion trick**: In the table, the distance-based weight splitting operation involves: (1) finding the distance vector from each point to each line, (2) calculating the distance vectors' norms, and (3) applying softmax over all lines in each tree. The first step costs $O(Lknd)$ computation and $O(Lknd)$ memory. Similarly, the second step costs $O(Lkdn)$ computation and $O(Lknd)$ memory.

Lastly, the final step costs $O(Lkn)$ computation and $O(Lkn)$ memory. Therefore, this operation theoretically costs $O(Lknd)$ in terms of both computation and memory. However, we leverage the automatic kernel fusion technique provided by torch.compile (Torch document) to *"fuse"* all these three steps into one single big step, thus contextualizing the distance vectors of size $Lkn \times d$ in a shared GPU memory instead of global memory. In the end, we only need to store two input matrices of size $Lk \times d$ (a line matrix) and $n \times d$ (a support matrix), along with one output matrix of size $Ln \times k$ (a split weight), which helps reduce the memory stored in GPU global memory to only $O(Lkn + Lkd + nd)$.

### C.7 Runtime and memory analysis

In this section, we have conducted a runtime comparison of Db-TSW with respect to the number of supports and the support's dimension in a single Nvidia A100 GPU. We fix $L = 100$ and $k = 10$ for all settings and varies $N \in \{500, 1000, 5000, 10000, 50000\}$ and $d \in \{10, 50, 100, 500, 1000\}$.

**Runtime evolution.** In Figure 7a, the relationship between runtime and number of supports demonstrates predominantly linear scaling, particularly beyond 10000 supports, though there's subtle non-linear behavior in the early portions of the curves for higher dimensions. The performance gap between high and low-dimensional cases widens as the number of supports increases, with $d = 1000$ taking approximately twice the runtime of $d = 500$, suggesting a linear relationship between dimension and computational time.

**Memory evolution.** In Figure 7b the memory consumption analysis of Db-TWD reveals several key patterns which align with the theoretical complexity analysis and suggest predictable scaling behavior. Firstly, there is a clear linear relationship between memory usage and the number of samples across all dimensions. Secondly, in higher-dimensional cases ($d = 100, 500, 1000$), the memory requirements exhibit a proportional relationship with the dimension, as evidenced by the approximately equal spacing between these curves. Finally, for lower dimensions ($d = 10, 50, 100$), the memory curves nearly overlap due to the dominance of the $Lkn$ term in the memory complexity formula $O(Lkn + Lkd + nd)$, where the number of supports components ($Lkn$) has a greater impact than the dimensional components ($Lkd$ and $nd$) when $L = 100$ and $k = 10$.

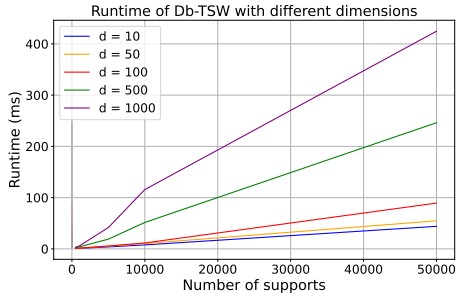
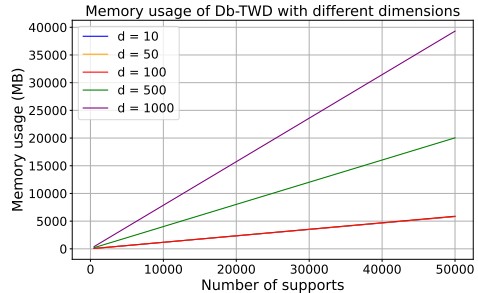

(a) Runtime evolution of Db-TSW with respect to different dimensions

(b) Memory evolution of Db-TSW with respect to different dimensions

Figure 7: Runtime and memory evolution of Db-TSW with respect to different dimension.

### C.8 Ablation study for $L$, $k$ and $\delta$

In this section, we include the ablation study for $L$, $k$, and $\delta$ on the Gradient flow task. We chose the Gaussian 100d dataset. When not chosen to vary, we used the default values: number of projections $n = 100$, learning rate $lr = 0.005$, number of iterations $n_{iter} = 5000$, number of trees $L = 500$, number of lines per tree $k = 2$, and coefficient $\delta = 10$. We fixed all other values while varying $k \in \{2, 4, 16, 64, 128\}$, $L \in \{100, 500, 1000, 1500, 2000, 3000\}$, and $\delta \in \{1, 2, 5, 10, 20, 50, 100\}$.

**Ablation study for $k$.** From Figure 8a, the ablation analysis of the number of lines $k$ reveals an interesting relationship between the number of lines per tree ($k$) and the algorithm's performance. When fixing the number of trees ($L$), configurations with fewer lines per tree demonstrate faster

convergence to lower Wasserstein distances. This behavior is reasonable because increasing $k$ expands the space of tree systems, which in turn requires more samples to attain similar performance levels. For instance, $k = 2$ and $k = 4$ configurations reach lower Wasserstein distances compared to $k = 64$ and $k = 128$. Additionally, as demonstrated in the runtime and memory analysis section C.7, configurations with higher number of lines per tree incur greater computational and memory costs while yielding suboptimal performance.

**Ablation study for** $L$. From Figure 8b, the convergence analysis demonstrates a clear relationship between the number of trees ($L$) and the algorithm's performance. When fixing the number of lines per tree ($k$), configurations with more trees achieve significantly better convergence, reaching much lower Wasserstein distances. This behavior is expected because increased sampling in the Monte Carlo method used in Db-TSW leads to a more accurate approximation of the distance. Specifically, comparing $L = 3000$ and $L = 100$, we observe that $L = 3000$ achieves a final Wasserstein distance of $10^{-5}$, while $L = 100$ only reaches $10^2$. However, there is a computation-accuracy trade-off since more trees involves more computation and memory.

**Ablation study for** $\delta$. From Figure 8c, the analysis of different $\delta$ values indicates variations in the algorithm's performance. Moderate values of delta ($\delta = 1 - 10$) enhance the convergence speed compared to $\delta = 1$, as they effectively accelerate the optimization process. However, when delta becomes too large ($\delta = 20 - 100$), the converged speed decreased.

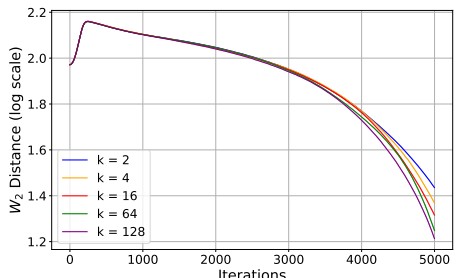
(a) Ablation study for the number of lines $k$.

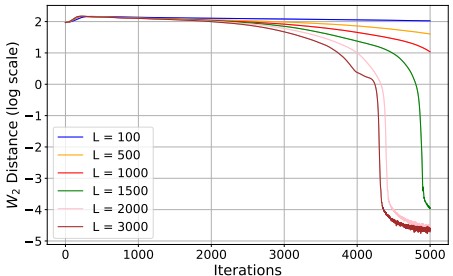
(b) Ablation study for the number of trees $L$.

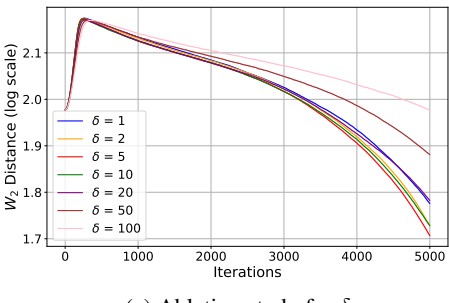
(c) Ablation study for $\delta$.

Figure 8: Ablation study for $k$, $L$, and $\delta$ on Gaussian 100d dataset.

