# OpenReview forum: "Distance-Based Tree-Sliced Wasserstein Distance"
_ICLR.cc/2025/Conference — ICLR 2025 Poster_

### Official Review · Reviewer_YmcA · 2024-10-27

**Soundness:** 4
**Presentation:** 4
**Contribution:** 3
**Rating:** 6
**Confidence:** 4

**Summary:**

This work proposes a variant of the TSW-SL distance. It is observed that the latter is not invariant w.r.t translations and rotations (Euclidean transformations), and that the splitting map used in TSW-SL does not take into account the trees but only the positions of the samples. Thus, the authors propose to take these two observations into account in order to define a new projection (Radon transform) on trees with a splitting map which is invariant w.r.t translation and rotations, and which takes into account the trees. Their new Radon transform is shown to be injective provided that the splitting map is invariant w.r.t Euclidean transformation. Then, they propose a splitting map which satisfies this, as well as two new more efficient ways to sample trees. Finally, they validate their new distance on several tasks such as diffusion models, gradient flows and color transfer.

**Strengths:**

This paper first identifies some drawbacks of the TSW-SL distance, and proposes to overcome them with a splitting map which depends on the trees sampled and which is Euclidean invariant. These new choices are well motivated and make sense.

Then, the authors provide the new Radon transform to project measures on trees using these splitting maps. They show that it is injective, and that the counterpart TSW distance is well an invariant distance. They also introduce a suitable splitting map, which they use in practice, and propose a new way to sample trees.

Finally, they experiment on challenging tasks such as generative modeling on high dimensional images with Diffusion, and demonstrate that their distance outperforms all previous Sliced-Wasserstein variants.

**Weaknesses:**

There is no statistical analysis of the new method. One could wonder how the distance evoluates w.r.t. the number of samples or the number of projections.

Another weakness is the lack of an ablation study, which would help us to see what really make it better than TSW-SL. Is it the new way of sampling trees which makes a big difference? Or is it because the distance is $E(d)$-equivariant? Or is it because the splitting map depends now on the distance to the lines?

**Questions:**

In equation (2), $Rf_\mu$ is not a probability. So I think there is an abuse of notation. In equation (8), $R_\mathcal{L}^\alpha\mu$ is used, but $R_\mathcal{L}^\alpha$ is only defined on $f\in L^2$ in Equation (7).

In Section 5.2, when suggesting to sample orthogonal directions, it is stated that it is "to ensure that the sampled tree systems do not include similar directions". While I understand what is meant here; the goal is to have different directions to better capture the topology of the data; the sentence is not very clear as in practice, we would have $\theta_i\neq \theta_h$ almost surely. Note also that it was proposed e.g. in [1] to sample orthogonal directions for Sliced-Wasserstein and in [2] for a max-Sliced-Wasserstein variant, but these references are missing.

I do not think I saw the step size used for each variant of SW in the gradient flow experiment. I believe that using the same step size might not always be fair as they do not necessarily have the same scale/smoothness properties (although I would agree that it is complicated to use a different step size for each method).


Typos:
- In legend of Figure 2, the two last sentences are almost the sames.
- Line 371: "$\mathcal{L}$ is a tree system consists"
- Line 476: "Db-TSW and $\mathrm{Db-TSW}^\top$ is"

[1] Rowland, M., Hron, J., Tang, Y., Choromanski, K., Sarlos, T., & Weller, A. (2019, April). Orthogonal estimation of Wasserstein distances. In The 22nd International Conference on Artificial Intelligence and Statistics (pp. 186-195). PMLR.

[2] Dai, B., & Seljak, U. (2020). Sliced iterative normalizing flows. arXiv preprint arXiv:2007.00674.

---

> ### Author Response · Authors · 2024-11-21
>
> We appreciate the reviewer’s feedback and have provided the following responses to address the concerns raised about our paper. Below, we summarize the weaknesses and questions highlighted by the reviewer and provide our answers accordingly.
>
> ---
>
> **W1+W2. There is no statistical analysis of the new method. One could wonder how the distance evaluates w.r.t. the number of samples or the number of projections.**
>
> **Another weakness is the lack of an ablation study, which would help us to see what really make it better than TSW-SL. Is it the new way of sampling trees which makes a big difference? Or is it because the distance is E(d)-equivariant? Or is it because the splitting map depends now on the distance to the lines?**
>
> **Answer.** We present an ablation study on the number of tree systems $L$ and the number of lines $k$ within each tree system.
>
> In Figure 8 in Appendix C.8, we include the ablation study for $L$ and $k$ on the Gradient flow task. We chose the Gaussian 100d dataset. When not chosen to vary, we used the default values: number of projections $n=100$, learning rate $lr=0.005$, number of iterations $n_{iter}=5000$, number of trees $L=500$, number of lines per tree $k=2$, and coefficient $\delta=10$. We fixed all other values while varying $k\in \{2,4,16,64,128\}$ and $L \in \{100, 500, 1000, 1500, 2000, 3000\}$.
>
> From Figure 8a in Appendix C.8, the ablation analysis of the number of lines $k$ reveals an interesting relationship between the number of lines per tree ($k$) and the algorithm's performance. When fixing the number of trees ($L$), configurations with fewer lines per tree demonstrate faster convergence to lower Wasserstein distances. This behavior is reasonable because increasing $k$ expands the space of tree systems, which in turn requires more samples to attain similar performance levels. For instance, $k=2$ and $k=4$ configurations reach lower Wasserstein distances compared to $k=64$ and $k=128$. Additionally, as demonstrated in the runtime and memory analysis (Appendix C.7), configurations with higher number of lines per tree incur greater computational and memory costs while yielding suboptimal performance.
>
> From Figure 8b in Appendix C.8, the convergence analysis demonstrates a clear relationship between the number of trees ($L$) and the algorithm's performance. When fixing the number of lines per tree ($k$), configurations with more trees achieve significantly better convergence, reaching much lower Wasserstein distances. This behavior is expected because increased sampling in the Monte Carlo method used in Db-TSW leads to a more accurate approximation of the distance. Specifically, comparing $L=3000$ and $L=100$, we observe that $L=3000$ achieves a final Wasserstein distance of $10^{-5}$, while $L=100$ only reaches $10^2$. However, there is a computation-accuracy trade-off since more trees involves more computation and memory.
>
> We believe that a comprehensive investigation into the analytical and statistical properties of Db-TSW is essential to theoretically explain these findings. However, as this paper primarily focuses on analyzing distance-based designs for splitting maps and the corresponding Radon Transform, and given the already broad scope of the work, we have opted to leave these aspects of Db-TSW for future research.
>
> **Q1. In equation (2), $Rf_{\mu}$ is not a probability. So I think there is an abuse of notation. In equation (8), $R_{\mathcal{L}}^{\alpha}\mu$ is used, but $R_{\mathcal{L}}^{\alpha}$ is only defined on $f \in L^2$ in equation (7).**
>
> **Answer.** We thank the reviewer for the correction. There is an abuse of notation between probability distributions and measures in the paper. We have made necessary adjustments in the revision.

---

> ### Author Response · Authors · 2024-11-21
>
> **Q2. In Section 5.2, when suggesting to sample orthogonal directions, it is stated that it is "to ensure that the sampled tree systems do not include similar directions". While I understand what is meant here; the goal is to have different directions to better capture the topology of the data; the sentence is not very clear as in practice, we would have $\theta_i \not= \theta_j$ almost surely. Note also that it was proposed e.g. in [1] to sample orthogonal directions for Sliced-Wasserstein and in [2] for a max-Sliced-Wasserstein variant, but these references are missing.**
>
> **Answer.** We recall the sentence mentioned by the reviewer in Section 5.2, located on line 382.
>
> > The intuitive motivation for this choice is to ensure that the sampled tree systems do not include lines with similar directions.
>
> Here, "similar" refers to "almost identical." Specifically, in a tree system $\{(x,\theta_1), \ldots, (x,\theta_k)\}$, we require $| \langle \theta_i, \theta_j \rangle | \ll 1$ to ensure that no two lines have nearly the same direction. This requirement motivates the introduction of the orthogonality condition in the space $\mathbb{T}^\perp$.**
>
> **Q3. I do not think I saw the step size used for each variant of SW in the gradient flow experiment. I believe that using the same step size might not always be fair as they do not necessarily have the same scale/smoothness properties (although I would agree that it is complicated to use a different step size for each method).**
>
> **Answer.** In the Gradient Flow experiment, we adopt the training settings, including the step size, from [1] and [2] for the existing methods and tune the step size for our approach. Since the step sizes for the existing methods were already optimized in the referenced papers, and we did not modify the settings or training process, we believe this ensures a fair comparison.
>
>
> ---
>
> **Reference.**
>
> [1] Guillaume Mahey et al., Fast optimal transport through sliced generalized wasserstein geodesics. In Advances in Neural Information Processing Systems 36 (2024).
>
> [2] Khai Nguyen et al., Sliced wasserstein estimation with control variates. In The Twelfth International Conference on Learning Representations, 2024.
>
> ---
>
> We sincerely thank the reviewer for the valuable feedback. If our responses adequately address all the concerns raised, we kindly hope the reviewer will consider raising the score of our paper.

---

> ### Author Response · Authors · 2024-11-22
> **Any Questions from Reviewer YmcA on Our Rebuttal?**
>
> We would like to thank the reviewer again for your thoughtful reviews and valuable feedback.
>
> We would appreciate it if you could let us know if our responses have addressed your concerns and whether you still have any other questions about our rebuttal.
>
> We would be happy to do any follow-up discussion or address any additional comments.

---

> > ### Comment · Reviewer_YmcA · 2024-11-23
> >
> > I would like to thank the authors for answering my review and revising the paper.
> >
> > Most of my concerns are resolved. Nonetheless, while the ablation study is a nice addition, I am still wondering which ingredient of the method makes the improvement over TSW-SL. I would have liked an additional experiment showing the difference with TSW-SL when e.g. using the new way of sampling trees instead of sampling chains, or sampling a chain and changing the splitting map. And I do not think the current ablation study answers these questions.
> >
> > Otherwise, I am satisfied with this work, and will leave my score unchanged.

---

> ### Author Response · Authors · 2024-11-23
> **Thanks for your endorsement!**
>
> Thank you for your response and additional suggestion. We appreciate your support and will include an ablation study exploring the effects of changing the sampling tree and modifying the splitting maps soon.

---

### Official Review · Reviewer_vezg · 2024-10-30

**Soundness:** 3
**Presentation:** 3
**Contribution:** 2
**Rating:** 6
**Confidence:** 3

**Summary:**

Paper proposes aims at enhancing the Tree-Sliced Wasserstein on a System of Line (TSW-SL) by defining a specific splitting maps (that are parts of the radon transform on systems of lines) that takes into account the distance to the lines. The method is coined Dd-TSW and it can be shown that it preserves some invariances (that are also preserved in $W_2$ or sliced-Wasserstein). By construction, it allows taking into account the distance to the lines and provides varying mass distributions. A variant that samples orthogonal lines, allowing to better cover the space, is also provided. Experiments show improved performances in several settings with similar computational timings.


**Note**: I have also reviewed the paper that introduces TSW-SL.

**Strengths:**

Dd-TSW is an extension of TSW-SL, in which the splitting map is refined and a more efficient method for sampling lines is provided. As such, the originality is fair. The definition of Dd-TSW is well-grounded and the experimental section shows that it yields competitive results in several scenario, even in one specific challenging scenario of high dimensional setting (GF experiment in 20d).

**Weaknesses:**

The main weaknesses of the paper lie in:

- Unsufficient analysis of its various parameters. The settings of some parameters are not discussed (e.g., the number of lines $k$ and the value of the inverse temperature parameter $\delta$).
- In the gradient flow experiment, differences in ground costs (e.g., $L_2$​ and tree metric) between methods make it challenging to compare results at a fixed number of iterations for a fixed LR.
- Competitors are not consistent between the different experiments.

**Questions:**

- The trees are recommended to be sampled close to the means of the target distributions. Does this relate to numerical instabilities that can occur for large values of $\delta$, such as those considered in the paper ($ \delta = 10$), to avoid excessively large distances?
- The $\delta = 10$ parameterization should lead to a very sparse splitting map. Could you provide an ablation study on this parameter and discuss why large values should be considered?
- Does the new method of sampling lines affect the performance of Dd-TSW, or is its definition justified only by computational issues?
- It seems (on the Gaussian 20D dataset) that Dd-TSW performs better in higher-dimensional cases than other SW variants. Is this result also true in higher-dimensional settings, and can you provide an explanation for this interesting behavior?
- It is stated in the introduction that Dd-TSW enables a highly parallelizable implementation, and in Section 6, it mentions "*that enables an algorithm with linear runtime and high parallelizability, keeping the wall-clock time of Dd-TSW and Dd-TSW⊥ approximately equal to that of vanilla SW and surpassing some SW variants*." Do the reported timings in section 6 relate to the parallel implementation?
- In the conclusion, it is mentioned that improved performance is expected by adapting recent advanced sampling schemes or techniques from SW to TSW-SL. However, it seems that the extension is not straightforward: can you provide concrete examples that could be adapted to Dd-TSW?

---

> ### Author Response · Authors · 2024-11-21
>
> We appreciate the reviewer’s feedback and have provided the following responses to address the concerns raised about our paper. Below, we summarize the weaknesses and questions highlighted by the reviewer and provide our answers accordingly.
>
> ---
>
> **W1. Unsufficient analysis of its various parameters. The settings of some parameters are not discussed (e.g., the number of lines $k$ and the value of the inverse temperature parameter $\delta$).**
>
> **Answer.** We appreciate the reviewer's suggestion about the analysis for hyperparameters of Db-TSW. In Figure 8 in Appendix C.8, we include the ablation study for $L$, $k$, and $\delta$ on the Gradient Flow task. We chose the Gaussian 100d dataset. When not chosen to vary, we used the default values: number of projections $n=100$, learning rate $lr=0.005$, number of iterations $n_{iter}=5000$, number of trees $L=500$, number of lines per tree $k=2$, and coefficient $\delta=10$. We fixed all other values while varying $k\in \{2,4,16,64,128\}$, $L \in \{100, 500, 1000, 1500, 2000, 3000\}$, and $\delta \in \{1, 2, 5, 10, 20, 50, 100\}$.
>
>
> From Figure 8a in Appendix C.8, the ablation analysis of the number of lines $k$ reveals an interesting relationship between the number of lines per tree ($k$) and the algorithm's performance. When fixing the number of trees ($L$), configurations with fewer lines per tree demonstrate faster convergence to lower Wasserstein distances. This behavior is reasonable because increasing $k$ expands the space of tree systems, which in turn requires more samples to attain similar performance levels. For instance, $k=2$ and $k=4$ configurations reach lower Wasserstein distances compared to $k=64$ and $k=128$. Additionally, as demonstrated in the runtime and memory analysis (Appendix C.7), configurations with higher number of lines per tree incur greater computational and memory costs while yielding suboptimal performance.
>
> From Figure 8b in Appendix C.8, the convergence analysis demonstrates a clear relationship between the number of trees ($L$) and the algorithm's performance. When fixing the number of lines per tree ($k$), configurations with more trees achieve significantly better convergence, reaching much lower Wasserstein distances. This behavior is expected because increased sampling in the Monte Carlo method used in Db-TSW leads to a more accurate approximation of the distance. Specifically, comparing $L=3000$ and $L=100$, we observe that $L=3000$ achieves a final Wasserstein distance of $10^{-5}$, while $L=100$ only reaches $10^2$. However, there is a computation-accuracy trade-off since more trees involves more computation and memory.
>
> From Figure 8c in Appendix C.8, the analysis of different $\delta$ values indicates variations in the algorithm's performance. Moderate values of delta ($\delta=1-10$) enhance the convergence speed compared to $\delta = 1$, as they effectively accelerate the optimization process.  However, when delta becomes too large ($\delta=20-100$), the converged speed decreased.
>
> **W2. In the gradient flow experiment, differences in ground costs (e.g., $L_2$ and tree metric) between methods make it challenging to compare results at a fixed number of iterations for a fixed LR.**
>
> **Answer.** As described in Section 6.2 of the paper (Lines 459–466), the objective of this task is to update the source distribution to make it as close to the target distribution as possible. The empirical advantage of Db-TSW over other SW-variants and over the original TSW-SL is demonstrated by measuring the $2$-Wasserstein distance between the source and target distributions at steps 500, 1000, 1500, 2000, and 2500. In this task:
>
> - Db-TSW and baselines are used as the loss function.
> - The $2$-Wasserstein distance serves as the evaluation metric.
>
> It is crucial to note that the roles of the $2$-Wasserstein distance (evaluation metric) and the other distances, such as Db-TSW, TSW-SL, and SW-variants (loss functions), are separate. Hence, there is no difference when evaluating the distance between the updated source distribution and the target distribution.

---

> > ### Author Response · Authors · 2024-11-21
> >
> > **W3. Competitors are not consistent between the different experiments.**
> >
> > **Answer.** In each experiment, we compare our methods with the baselines provided in the most recent papers that includes the corresponding task. Empirical evidence demonstrates that Db-TSW consistently outperforms state-of-the-art SW variants across various tasks. For instance, in the Denoising Diffusion Model task, we include all baselines from the latest relevant paper [2]. As shown in Table 1 (Section 6.1), [2] reports the best performance with an FID of $2.70$, while our method achieves FIDs of $2.60$ and $2.525$. We believe this effectively highlights the advantages of Db-TSW.
> >
> > **Q1. The trees are recommended to be sampled close to the means of the target distributions. Does this relate to numerical instabilities that can occur for large values of $\delta$, such as those considered in the paper ($\delta = 10$), to avoid excessively large distances?**
> >
> > **Answer**: Through empirical observations, we discovered that initializing the tree root near the mean of the target distribution results in more consistent performance for our models. Specifically, if $m$ represents the mean position of all data points, the root $r$ is sampled uniformly from a small closed ball centered at $m$. This was mentioned in line 421 of our paper.
> >
> > > This allows for more effective sampling, specifically by sampling trees located around the support of target distributions (ideally near their means).
> >
> > We believe that a comprehensive investigation into the analytical and statistical properties of Db-TSW is essential to theoretically explain these findings. However, as this paper primarily focuses on analyzing distance-based designs for splitting maps and the corresponding Radon Transform, and given the already broad scope of the work, we have opted to leave these aspects of Db-TSW for future research.
> >
> > **Q2. The $\delta = 10$ parameterization should lead to a very sparse splitting map. Could you provide an ablation study on this parameter and discuss why larger values should not be considered?**
> >
> > **Answer.** From Figure 8c in Appendix C.8, the analysis of different $\delta$ values indicates variations in the algorithm's performance. Moderate values of delta ($\delta=1-10$) enhance the convergence speed compared to $\delta = 1$, as they effectively accelerate the optimization process. However, when delta becomes too large ($\delta=20-100$), the converged speed decreased.
> >
> > **Q3. Does the new method of sampling lines affect the performance of Dd-TSW, or is its definition justified only by computational issues?**
> >
> > **Answer.** Compared to the original TSW-SL, our new splitting maps are also applicable to all tree structures because their definition is based solely on the positional information of lines within tree systems, rather than the specific structure of the trees. Additionally, the new class of splitting maps is significantly broader than the previous one, which we believe is sufficient to enhance the performance of the corresponding Db-TSW distance. Empirical results presented in the experimental section demonstrate that Db-TSW outperforms the original TSW-SL and other recent SW variants. We opted to showcase the concurrent-line tree structure because it enables a more efficient implementation compared to other tree structures.

---

> > > ### Author Response · Authors · 2024-11-21
> > >
> > > **Q3. Does the new method of sampling lines affect the performance of Dd-TSW, or is its definition justified only by computational issues?**
> > >
> > > **Answer.** Compared to the original TSW-SL, our new splitting maps are also applicable to all tree structures because their definition is based solely on the positional information of lines within tree systems, rather than the specific structure of the trees. Additionally, the new class of splitting maps is significantly broader than the previous one, which we believe is sufficient to enhance the performance of the corresponding Db-TSW distance. Empirical results presented in the experimental section demonstrate that Db-TSW outperforms the original TSW-SL and other recent SW variants. We opted to showcase the concurrent-line tree structure because it enables a more efficient implementation compared to other tree structures.
> > >
> > > **Q4. It seems on the Gaussian 20D dataset that Db-TSW performs better in higher-dimensional cases than other SW variants. Is this result also true in higher-dimensional settings, and can you provide an explanation for this interesting behavior?**
> > >
> > > **Answer.** The motivation for this paper arises from a simple yet intriguing idea: In the framework of Sliced Wasserstein (SW), a probability distribution on $\mathbb{R}^d$ is pushed forward onto a line. This raises the question: what does the resulting distribution reveal about the original one? It is evident that distinct distributions, when projected onto the same line, can become indistinguishable.
> > >
> > > Now, let us compare a tree system composed of two lines, $a$ and $b$ in $\mathbb{R}^2$, along with a distribution $\mu$ on $\mathbb{R}^2$. For simplicity, assume that $\mu$ is a Dirac delta distribution.
> > >
> > > - Many Dirac delta distributions in $\mathbb{R}^2$ become identical after being projected onto line $a$, and the same holds true for line $b$. However, in most cases, the projections of $\mu$ on both lines of the tree system can uniquely identify the original Dirac delta distribution. ("Most cases" excludes exceptional scenarios, such as when $a$ and $b$ are the same line.)
> > > - The above reasoning might appear insufficient because it evaluates the tree system by considering only one of its lines at a time. What happens if we evaluate the tree system using both lines together? This is where the splitting map becomes essential. The splitting map enables a more versatile allocation of mass between the two lines, rather than concentrating all the mass onto one line.
> > >
> > > We believe this intuitive explanation adequately addresses the reviewer's concern regarding higher-dimensional cases. In our experiments, the Denoising Diffusion Model task serves as a high-dimensional test case, with a dimensionality of $32 \times 32 \times 3$, where $32 \times 32$ is the size of pictures and $3$ is the RGB dimension. Empirical results demonstrate that Db-TSW consistently outperforms SW variants in such high-dimensional tasks, further validating its effectiveness.
> > >
> > > **Q5. It is stated in the introduction that Dd-TSW enables a highly parallelizable implementation, and in Section 6, it mentions "that enables an algorithm with linear runtime and high parallelizability, keeping the wall-clock time of Dd-TSW and Dd-TSW$^\perp$ approximately equal to that of vanilla SW and surpassing some SW variants." Do the reported timings in section 6 related to the parallel implementation?**
> > >
> > > **Answer.** The reported timings in Section 6 are based on a parallel implementation executed on a single A100 GPU. We also provide a detailed discussion in Appendix C.7 how the runtime of Db-TSW scales with the number of supports $n$ and the dimensionality of the supports $d$. Additionally, we address memory limitations by employing the kernel fusion trick in Appendix C.6.

---

> ### Author Response · Authors · 2024-11-21
>
> **Q6. In the conclusion, it is mentioned that improved performance is expected by adapting recent advanced sampling schemes or techniques from SW to TSW-SL. However, it seems that the extension is not straightforward: can you provide some examples that could be adapted to Dd-TSW?**
>
> **Answer.** The tree-sliced framework in our paper has the following local perspective: Each line in a tree system is treated similarly to a line in the Sliced Wasserstein (SW) framework. Splitting maps determine how the mass at each point is distributed across the lines, and then the projection of these mass portions onto the lines is processed in the same way as in SW. A significant challenge with this approach is to verify whether the injectivity of the corresponding Radon Transform is preserved, as this determines whether the proposed metric qualifies as a true metric or merely a pseudo-metric. However, we addressed this concern by providing the proof in Appendix B.3of the paper. Considering, for example, the Generalized Sliced Wasserstein (GSW) distance [1], in GSW, the SW framework is retained, but the projection mechanism is altered. Specifically, GSW generalizes the integration level set in the Radon Transform, replacing the level set defined by the inner product (representing orthogonal hyperplanes) with one defined by an arbitrary function. Similarly, in TSW-SL, which currently relies on the inner product, a framework could be developed that generalizes this by using an arbitrary function, offering new flexibility and applications.
>
> ---
>
> **Reference.**
>
> [1] Soheil Kolouri et al., Generalized Sliced Wasserstein Distances. Advances in neural information processing systems, 32, 2019
>
> [2] Khai Nguyen et al., Sliced wasserstein with random-path projecting directions. In Forty-first International Conference on Machine Learning, 2024
>
> ---
>
> We sincerely thank the reviewer for the valuable feedback. If our responses adequately address all the concerns raised, we kindly hope the reviewer will consider raising the score of our paper.

---

> ### Author Response · Authors · 2024-11-22
> **Any Questions from Reviewer vezg on Our Rebuttal?**
>
> We would like to thank the reviewer again for your thoughtful reviews and valuable feedback.
>
> We would appreciate it if you could let us know if our responses have addressed your concerns and whether you still have any other questions about our rebuttal.
>
> We would be happy to do any follow-up discussion or address any additional comments.

---

> > ### Comment · Reviewer_vezg · 2024-11-25
> >
> > Thanks for your detailed answers and additional results.
> >
> > The main justification for my score was the good behaviour of the method in high dimensions, as shown in the experiment for GFs in dimension 20. In view of the ablation study added in the appendix, it seems that this is not so obvious for dimension 100, unless we consider a large number of trees, which may be in contradiction with the idea of having a computationally interesting method.
> > Minor comment: I'm surprised by the SWGG results in dimension 20, which converge less quickly than for SW in dimension 20, whereas in a similar experiment (in dimension 500) in the paper, the opposite phenomenon was observed.
> >
> > So I'm keeping my score unchanged

---

> ### Author Response · Authors · 2024-11-25
> **Thanks for your endorsement!**
>
> Thank you for your response, and we appreciate your endorsement. For high-dimensional cases, advanced methods such as SWGG may require a more suitable number of Monte Carlo samples to fully realize their advantages over the original SW. The behaviors of SW and TSW variants in high dimensions are indeed intriguing problems, and we hope that more techniques related to SW and TSW will be discovered in the future.

---

### Official Review · Reviewer_2G1e · 2024-10-31

**Soundness:** 3
**Presentation:** 3
**Contribution:** 2
**Rating:** 6
**Confidence:** 4

**Summary:**

The paper introduces a distance-based variant of the tree-sliced SW on a system of lines (TSW-SL) [1]. The core contribution lies in generalizing the class of splitting maps used to project measures onto tree systems while preserving Euclidean invariance. The authors provide theoretical results for their method, including proofs of metric properties and injectivity of the associated Radon transform. They also introduce a simplified tree sampling process that with efficient GPU implementation. Various experiments including gradient flows, image generation, and color transfer show competitive or better performance in certain hyperparameter settings.

[1] Tran, Viet-Hoang, et al. "Tree-Sliced Wasserstein Distance on a System of Lines." arXiv preprint arXiv:2406.13725 (2024).

**Strengths:**

S1: Generalization of splitting maps addresses a limitation in previous work [1].

S2: Efficient implementation thanks to the proposed simplified tree sampling process.

S3: The writing is well-written and easy-to-follow.

S4: Good empirical results within the given setups.

**Weaknesses:**

W1: It would be nice to have an explicit discussion of computational/memory complexity.

W2: For setups that aggregate results from multiple runs (e.g., gradient flow), the paper should include both mean and std/variance (instead of just the means). Additionally, it may be a good idea to use plots to better present the results in Table 2,3 (visually). Both of these are good practices in general, and also used in [2], which is cited in the paper.

W3: The paper would benefit from a detailed discussion on various instances of tree systems and especially splitting maps (which are directly related to the primary contributions). For instance, using $k$ concurrent lines with shared root is fast but what are we sacrificing? Are they quantifiable? Softmax works well as an E(d)-invariant splitting map but what are some other instances (along with pros and cons)?  Along those lines, a stronger ablation section would be helpful. It should clearly demonstrate that any performance gain is due to the novel technical contributions to support the paper's claims throughout (is it the invariance or the proposed tree structure that gives better performance?). How much is the gain if one adjusts $L,k, \delta$ for datasets of varying complexity? What about root sampling? Since the design space includes various components, that would help practitioners in making sensible design choices for their applications.

W4: The paper would be strengthened if explicitly discussing sample/projection complexity, and also error bounds, convergence rate for MC approximation.

W5: What are the authors' rationales for not using a consistent set of baseline methods to evaluate relative performance across tasks? Moreover, is it a good idea to compare against non-linear slicing methods like GSW, ASW? Why or why not?

---
[2] Mahey, Guillaume, et al. "Fast Optimal Transport through Sliced Generalized Wasserstein Geodesics." Advances in Neural Information Processing Systems 36 (2024).

[3] Kolouri, Soheil, et al. "Generalized sliced wasserstein distances." Advances in neural information processing systems 32 (2019).

[4] Chen, Xiongjie, Yongxin Yang, and Yunpeng Li. "Augmented sliced Wasserstein distances." arXiv preprint arXiv:2006.08812 (2020).

**Questions:**

Q1: Can any component of this framework be made learnable? That would probably enable various interesting extensions.

Q2: Are there instances where prior knowledge on the data allows a good/optimal tree design?

---

> ### Author Response · Authors · 2024-11-21
>
> We appreciate the reviewer’s feedback and have provided the following responses to address the concerns raised about our paper. Below, we summarize the weaknesses and questions highlighted by the reviewer and provide our answers accordingly.
>
> ---
>
> **W1. It would be nice to have an explicit discussion of computational/memory complexity.**
>
> **Answer.** Since GPUs are now standard in machine learning workflows due to their parallel processing capabilities, and Db-TSW is primarily intended use for deep learning tasks, like generative modeling and optimal transport problems, which are typically GPU-accelerated, we provide both complexity analysis and empirical runtime/memory of Db-TSW with GPU settings. We have also included these information in Appendix C.6 and C.7 of our revised manuscript.
>
> Assume $n \ge m$, the computational and memory complexity of Db-TSW are $O(Lknd + Lkn\log n)$ and $O(Lkn + Lkd + nd)$, respectively. In details, the Db-TSW algorithm will have three most costly operations as described in the table below (also Table 5 in Appendix C.6):
>
> *Table: Complexity Analysis of Db-TSW*
> | Operation | Description | Computation | Memory |
> |-----------|-------------|------------|---------|
> | Projection | Matrix multiplication of points and lines | $O(Lknd)$ | $O(Lkd + nd)$ |
> | Distance-based weight splitting | Distance calculation and softmax | $O(Lknd)$ | $O(Lkn + Lkd + nd)$ |
> | Sorting | Sorting projected coordinates | $O(Lkn\log n)$ | $O(Lkn)$ |
> | **Total** | | $O(Lknd + Lkn\log n)$ | $O(Lkn + Lkd + nd)$ |
>
> **The kernel fusion trick**: In the table above, the distance-based weight splitting operation involves: (1) finding the distance vector from each point to each line, (2) calculating the distance vectors' norms, and (3) applying softmax over all lines in each tree. The first step costs $O(Lknd)$ computation and $O(Lknd)$ memory. Similarly, the second step costs $O(Lkdn)$ computation and $O(Lknd)$ memory. Lastly, the final step costs $O(Lkn)$ computation and $O(Lkn)$ memory. Therefore, this operation theoretically costs $O(Lknd)$ in terms of both computation and memory. However, we leverage the automatic kernel fusion technique provided by `torch.compile` [(torch document)](https://pytorch.org/docs/stable/generated/torch.compile.html) to "fuse" all these three steps into one single big step, thus contextualizing the distance vectors of size $Lkn \times d$ in a shared GPU memory instead of global memory. In the end, we only need to store two input matrices of size $Lk\times d$ (a line matrix) and $n \times d$ (a support matrix), along with one output matrix of size $Ln \times k$ (a split weight), which helps reduce the memory stored in GPU global memory to only $O(Lkn + Lkd + nd)$.
>
> In Appendix C.7, we have conducted a runtime and memory comparison of Db-TSW with respect to the number of supports and the support's dimension in a single Nvidia A100 GPU. We fix $L=100$ and $k=10$ for all settings and varies $N \in \{500, 1000, 5000, 10000, 50000\}$ and $d \in \{10, 50, 100, 500, 1000\}$.
>
> In Figure 7a (Appendix C.7), the relationship between runtime and number of supports demonstrates predominantly linear scaling, particularly beyond $10000$ supports, though there is subtle non-linear behavior in the early portions of the curves for higher dimensions. The performance gap between high and low-dimensional cases widens as the number of supports increases, with $d=1000$ taking approximately twice the runtime of $d=500$, suggesting a linear relationship between dimension and computational time.
>
> In Figure 7b (Appendix C.7), the memory consumption analysis of Db-TWD reveals several key patterns which align with the theoretical complexity analysis and suggest predictable scaling behavior. First, there is a clear linear relationship between memory usage and the number of samples across all dimensions. Second, in higher-dimensional cases ($d = 100, 500, 1000$), the memory requirements exhibit a proportional relationship with the dimension, as evidenced by the approximately equal spacing between these curves. Finally, for lower dimensions ($d = 10, 50, 100$), the memory curves nearly overlap due to the dominance of the $Lkn$ term in the memory complexity formula $O(Lkn + Lkd + nd)$, where the number of supports components ($Lkn$) has a greater impact than the dimensional components ($Lkd$ and $nd$) when $L=100$ and $k=10$.

---

> > ### Author Response · Authors · 2024-11-21
> >
> > **W2. For setups that aggregate results from multiple runs (e.g., gradient flow), the paper should include both mean and std/variance (instead of just the means). Additionally, it may be a good idea to use plots to better present the results in Table 2,3 (visually). Both of these are good practices in general, and also used in [2], which is cited in the paper.**
> >
> > **Answer.** We show the detail result of Db-TSW and Db-TSW$^\perp$ in Table 4 (Appendix C.3) and visualize the result in Figure 5 (Appendix C.3). Figure 5 reveals distinct performance patterns among different methods, with TSW variants showing faster convergence rate. Notably, Db-TSW$^\perp$ achieves the fastest convergence, followed by Db-TSW and TSW-SL, all exhibiting a sharp performance improvement around iteration 1500. While these methods show some fluctuation mid-process as indicated by the shaded variance regions, they maintain remarkable stability in the final iterations (itetaion 2000-2500). These results are further supported by the quantitative mean and standard deviation values provided in Table 4. Here, we include the table for easier observation. Note: **bold** indicates best performance, *italic* indicates second best.
> >
> > *Table: Wasserstein distance between source and target distributions of 3 runs on Gaussian 20d dataset.*
> > | Methods | Iteration |||||
> > |---|:---:|:---:|:---:|:---:|:---:|
> > | | 500 | 1000 | 1500 | 2000 | 2500 |
> > | SW | 17.57 ± 2.4e-1 | 15.86 ± 3.1e-1 | 13.92 ± 3.7e-1 | 11.70 ± 4.2e-1 | 9.22 ± 3.8e-1 |
> > | SWGG | 16.53 ± 1.2e-1 | 16.64 ± 1.4e-1 | 16.65 ± 1.7e-1 | 16.63 ± 1.6e-1 | 16.64 ± 1.5e-1 |
> > | LCVSW | 16.86 ± 4e-1 | 14.36 ± 3.4e-1 | 11.68 ± 3.3e-1 | 8.80 ± 4e-1 | 5.91 ± 2.3e-1 |
> > | TSW-SL | 12.61 ± 3e-1 | 5.68 ± 3.8e-1 | 6.90e-1 ± 8e-2 | 6.83e-4 ± 2.6e-5 | 4.22e-4 ± 1.2e-5 |
> > | Db-TSW | *12.46* ± 4.6e-1 | *5.12* ± 3.9e-1 | *4.8e-1* ± 3.3e-1 | *6.08e-4* ± 6.7e-5 | *3.84e-4* ± 2.2e-5 |
> > | Db-TSW⊥ | **12.15** ± 3.5e-1 | **4.67** ± 3.6e-1 | **1.22e-1** ± 1.6e-1 | **5.74e-4** ± 2e-5 | **3.78e-4** ± 1.4e-5 |

---

> > > ### Author Response · Authors · 2024-11-21
> > >
> > > **W3. The paper would benefit from a detailed discussion on various instances of tree systems and especially splitting maps (which are directly related to the primary contributions). For instance, using $k$ concurrent lines with shared root is fast but what are we sacrificing? Are they quantifiable? Softmax works well as an E(d)-invariant splitting map but what are some other instances (along with pros and cons)? ... How much is the gain if one adjusts $L,k,\delta$ for datasets of varying complexity? What about root sampling? Since the design space includes various components, that would help practitioners in making sensible design choices for their applications.**
> > >
> > > **Answer.** Thanks for for your comments. We address these concerns from the reviewer by responding to each point separately.
> > >
> > > >The paper would benefit from a detailed discussion on various instances of tree systems and especially splitting maps (which are directly related to the primary contributions). For instance, using $k$ concurrent lines with shared root is fast but what are we sacrificing? Are they quantifiable?
> > >
> > > Compared to the original TSW-SL, our new splitting maps are also applicable to all tree structures because their definition is based solely on the positional information of lines within tree systems, rather than the specific structure of the trees. Additionally, the new class of splitting maps is significantly broader than the previous one, which we believe is sufficient to enhance the performance of the corresponding Db-TSW distance. Empirical results presented in the experimental section demonstrate that Db-TSW outperforms the original TSW-SL and other recent SW variants. We opted to showcase the concurrent-line tree structure because it enables a more efficient implementation compared to other tree structures.
> > >
> > > > Along those lines, a stronger ablation section would be helpful. It should clearly demonstrate that any performance gain is due to the novel technical contributions to support the paper's claims throughout (is it the invariance or the proposed tree structure that gives better performance?) How much is the gain if one adjusts $L,k,\delta$ for datasets of varying complexity?
> > >
> > > In Figure 8 in Appendix C.8, we include the ablation study for $L$, $k$, and $\delta$ on the Gradient flow task. We chose the Gaussian 100d dataset. When not chosen to vary, we used the default values: number of projections $n=100$, learning rate $lr=0.005$, number of iterations $n_{iter}=5000$, number of trees $L=500$, number of lines per tree $k=2$, and coefficient $\delta=10$. We fixed all other values while varying $k\in \{2,4,16,64,128\}$, $L \in \{100, 500, 1000, 1500, 2000, 3000\}$, and $\delta \in \{1, 2, 5, 10, 20, 50, 100\}$.
> > >
> > >
> > > From Figure 8a in Appendix C.8, the ablation analysis of the number of lines $k$ reveals an interesting relationship between the number of lines per tree ($k$) and the algorithm's performance. When fixing the number of trees ($L$), configurations with fewer lines per tree demonstrate faster convergence to lower Wasserstein distances. This behavior is reasonable because increasing $k$ expands the space of tree systems, which in turn requires more samples to attain similar performance levels. For instance, $k=2$ and $k=4$ configurations reach lower Wasserstein distances compared to $k=64$ and $k=128$. Additionally, as demonstrated in the runtime and memory analysis (Appendix C.7), configurations with higher number of lines per tree incur greater computational and memory costs while yielding suboptimal performance.
> > >
> > > From Figure 8b in Appendix C.8, the convergence analysis demonstrates a clear relationship between the number of trees ($L$) and the algorithm's performance. When fixing the number of lines per tree ($k$), configurations with more trees achieve significantly better convergence, reaching much lower Wasserstein distances. This behavior is expected because increased sampling in the Monte Carlo method used in Db-TSW leads to a more accurate approximation of the distance. Specifically, comparing $L=3000$ and $L=100$, we observe that $L=3000$ achieves a final Wasserstein distance of $10^{-5}$, while $L=100$ only reaches $10^2$. However, there is a computation-accuracy trade-off since more trees involves more computation and memory.
> > >
> > > From Figure 8c in Appendix C.8, the analysis of different $\delta$ values indicates variations in the algorithm's performance. Moderate values of delta ($\delta=1-10$) enhance the convergence speed compared to $\delta = 1$, as they effectively accelerate the optimization process.  However, when delta becomes too large ($\delta=20-100$), the converged speed decreased.
> > >
> > >
> > > > What about root sampling?
> > >
> > > We answer this question in **Q2.** below.

---

> ### Author Response · Authors · 2024-11-21
>
> **W4. The paper would be strengthened if explicitly discussing sample/projection complexity, and also error bounds, convergence rate for MC approximation.**
>
> **Answer.** This paper emphasizes the analysis of both traditional and newly introduced splitting maps, particularly the $E(d)$-invariant splitting maps, in relation to the Radon Transform. Additionally, we propose the novel Db-TSW approach. Due to the extensive content presented, we have deferred the analytical and statistical examination of STSW to future work.
>
> It is important to note that the analysis of Db-TSW poses unique challenges arising from the inclusion of splitting maps—a distinctive feature of Tree-Sliced Wasserstein (TSW) variants that differentiates them from Sliced Wasserstein (SW) variants.
>
> We are actively investigating the properties of splitting maps, which appear to be a highly promising research direction.
>
> **W5. What are the authors' rationales for not using a consistent set of baseline methods to evaluate relative performance across tasks? Moreover, is it a good idea to compare against non-linear slicing methods like GSW, ASW? Why or why not?**
>
> **Answer.** In each experiment, we compare our methods with the baselines provided in the most recent papers that include the corresponding task. Empirical evidence demonstrates that Db-TSW consistently outperforms state-of-the-art SW variants across various tasks. For instance, in the Denoising Diffusion Model task, we include all baselines from the latest relevant paper [1]. As shown in Table 1, [1] reports the best performance with an FID of 2.70, while our method achieves FIDs of 2.60 and 2.525. We believe this effectively highlights the advantages of Db-TSW.
>
> **Q1. Can any component of this framework be made learnable? That would probably enable various interesting extensions.**
>
> **Answer.** We appreciate the reviewer’s suggestion. There are two straightforward approaches to incorporate a learnable perspective: (1) tuning the parameter $\delta$ and (2) modifying the splitting map $\alpha$.
>
> For $\delta$, it can be parameterized as a single learnable variable. For splitting maps, recall that the splitting maps used in our method are defined in Eq. (25):
>
> $\alpha(x,\mathcal{L})\_l = softmax \Bigl( \{\delta \cdot d(x,\mathcal{L})\_l\}\_{l \in \mathcal{L}} \Bigr)$.
>
> More generally, one can design splitting maps in the form shown in Eq. (24):
>
> $\alpha(x,\mathcal{L})\_l = \beta \Bigl( \{d(x,\mathcal{L})\_l\}\_{l \in \mathcal{L}} \Bigr)$,
>
> where $\beta \colon \mathbb{R}^k \to \Delta_{k-1}$ is an arbitrary map. As discussed in Section 5.2, this design ensures that the splitting maps remain $E(d)$-invariant. To make $\beta$ learnable, it can be parameterized as an MLP with input and output dimensions equal to $k$, using a softmax layer at the end to ensure the output lies on the standard simplex $\Delta_{k-1}$.
>
> Due to the limited time available during the discussion phase, we are unable to provide empirical results for this approach. However, we have conducted an ablation study on $\delta$. In Figure 8c in Appendix C.8, the analysis of different $\delta$ values indicates variations in the algorithm's performance. Moderate values of delta ($\delta=1-10$) enhance the convergence speed compared to $\delta = 1$, as they effectively accelerate the optimization process. However, when delta becomes too large ($\delta=20-100$), the converged speed decreased.
>
> **Q2. Are there instances where prior knowledge on the data allows a good/optimal tree design?**
>
> **Answer.** Currently, the only prior knowledge about the data used for tree design is that the root of concurrent-line tree systems is sampled near the data's mean. Specifically, if $m$ represents the mean position of all data points, the root $r$ is sampled uniformly from a small closed ball centered at $m$. This was mentioned in line 421 of our paper.
>
> > This allows for more effective sampling, specifically by sampling trees located around the support of target distributions (ideally near their means).
>
> Through empirical observations, we discover that initializing the tree root near the mean of the target distribution results in more consistent performance for our models. We believe that a comprehensive investigation into the analytical and statistical properties of Db-TSW is essential to theoretically explain these findings. However, as this paper primarily focuses on analyzing distance-based designs for splitting maps and the corresponding Radon Transform, and given the already broad scope of the work, we have opted to leave these aspects of Db-TSW for future research.
>
> ---
>
> **References**
>
> [1] Khai Nguyen et al., Sliced wasserstein with random-path projecting directions. ICML 2024
>
> ---
>
> We sincerely thank the reviewer for the valuable feedback. If our responses adequately address all the concerns raised, we kindly hope the reviewer will consider raising the score of our paper.

---

> ### Author Response · Authors · 2024-11-22
> **Any Questions from Reviewer 2G1e on Our Rebuttal?**
>
> We would like to thank the reviewer again for your thoughtful reviews and valuable feedback.
>
> We would appreciate it if you could let us know if our responses have addressed your concerns and whether you still have any other questions about our rebuttal.
>
> We would be happy to do any follow-up discussion or address any additional comments.

---

### Official Review · Reviewer_xP5V · 2024-11-04

**Soundness:** 3
**Presentation:** 3
**Contribution:** 3
**Rating:** 8
**Confidence:** 3

**Summary:**

The paper introduces the Distance-based Tree-Sliced Wasserstein (Db-TSW) distance, which addresses limitations in existing Tree-Sliced Wasserstein on Systems of Lines (TSW-SL) by improving Euclidean invariance and preserving positional information in optimal transport (OT). Unlike traditional Sliced Wasserstein (SW) distances, which can lose topological information, Db-TSW incorporates tree structures and a novel class of E(d)-invariant splitting maps to maintain metric consistency under Euclidean transformations. This approach also includes a new tree sampling method for efficiency, enabling GPU-friendly implementations. Through rigorous theoretical analysis and empirical experiments, Db-TSW is shown to outperform recent SW variants on tasks such as gradient flows, image style transfer, and generative modeling, while remaining computationally efficient​.

**Strengths:**

* Paper is very-well written and clear.
* Through extensive experiments, authors show the effectiveness of Db-TSW and compare it with existing benchmarks.
* Authors propose a novel variant of Radon transform on Systems of Lines which generalizes the invariant splitting mapping of the previous works.

**Weaknesses:**

* Although Db-TSW is optimized for GPU implementation, the method’s reliance on high-dimensional Euclidean distances for each point-line relationship could still pose challenges for scalability in very high-dimensional settings. This could result in increased computation times or memory usage for large datasets, particularly when fine-grained positional information is necessary. I would like to see a thorough runtime comparison of Db-TSW with respect to the number of samples and dimensionality of the data.

* The root-concurrent structure for sampling tree systems, though efficient, limits Db-TSW’s flexibility in capturing diverse spatial arrangements. I am curious about how effectively Db-TSW captures these spatial variations. Is there a way to demonstrate this? Could you provide an example where Db-TSW might fail to capture them?

**Questions:**

* The root-concurrent structure is computationally efficient, but it may affect the transform's injectivity, especially in cases with overlapping or closely aligned lines. How does this design ensure that unique positional information is retained across sampled trees, and are there any trade-offs in injectivity or stability in high-dimensional data?

* How does Db-TSW handle cases where the Euclidean distance between points and tree lines is nearly uniform across lines, potentially reducing the distinctiveness of the splitting map?

---

> ### Author Response · Authors · 2024-11-21
>
> We appreciate the reviewer’s feedback and have provided the following responses to address the concerns raised about our paper. Below, we summarize the weaknesses and questions highlighted by the reviewer and provide our answers accordingly.
>
> ---
>
>
> **W1. Although Db-TSW is optimized for GPU implementation, the method’s reliance on high-dimensional Euclidean distances for each point-line relationship could still pose challenges for scalability in very high-dimensional settings. This could result in increased computation times or memory usage for large datasets, particularly when fine-grained positional information is necessary. I would like to see a thorough runtime comparison of Db-TSW with respect to the number of samples and dimensionality of the data.**
>
> **Answer.** We agree with the reviewer that the point-line relationship theoretically introduces an additional memory and computation usage. Particularly, assume that $n \ge m$, the weight splitting operator will involve $O(Lkdn)$ memory and $O(Lkdn)$ computation to calculate distance vectors from all points to all lines. We overcome the $O(Lkdn)$ memory issue by using automatic kernel fusion feature of `torch.compile`. This technique allows us to fuse the distance vector calculating function and its norm calculating function into one function, thus only contextualize the distance vectors in the shared memory instead of in the global memory of a GPU. Meanwhile, the $O(Lkdn)$ computation of the weight splitting operator does not affect the overall computation complexity since it has the same computation complexity with the projection operator used widely in SW and its variants implementation. Both our weight splitting operator and the projection operator require a matrix multiplication for two matrixes of size $n \times d$ and $Lk\times d$. We have also included the detail of complexity analysis and runtime/memory evolution in Appendix C.6 and C.7 of the paper.
>
> In Appendix C.7 of our revision, we have conducted a runtime comparison of Db-TSW with respect to the number of supports and the support's dimension in a single Nvidia A100 GPU. We fix $L=100$ and $k=10$ for all settings and varies $N \in \{500, 1000, 5000, 10000, 50000\}$ and $d \in \{10, 50, 100, 500, 1000\}$.
>
> In Figure 7a (Appendix C.7), the relationship between runtime and number of supports demonstrates predominantly linear scaling, particularly beyond $10000$ supports, though there is subtle non-linear behavior in the early portions of the curves for higher dimensions. The performance gap between high and low-dimensional cases widens as the number of supports increases, with $d=1000$ taking approximately twice the runtime of $d=500$, suggesting a linear relationship between dimension and computational time.

---

> > ### Author Response · Authors · 2024-11-21
> >
> > **W2+Q1. The root-concurrent structure for sampling tree systems, though efficient, limits Db-TSW’s flexibility in capturing diverse spatial arrangements. I am curious about how effectively Db-TSW captures these spatial variations. Is there a way to demonstrate this? Could you provide an example where Db-TSW might fail to capture them?**
> >
> > **The root-concurrent structure is computationally efficient, but it may affect the transform's injectivity, especially in cases with overlapping or closely aligned lines. How does this design ensure that unique positional information is retained across sampled trees, and are there any trade-offs in injectivity or stability in high-dimensional data?**
> >
> > **Answer.** The root-concurrent structure still preserves injectivity, or more precisely, ensures that Db-TSW remains a metric. This is theoretically proven in Appendix B.3 and Remark 10 of our manuscript. Specifically, the proof in Appendix B.3 applies the Radon Transform, which considers all tree systems, while Remark 10 explains why the proof can be extended to a collection of tree systems with an additional property. We recall Remark 10 as follows.
> >
> > > *Remark 10.* The injectivity still hold if we restrict $\mathbb{L}^d_k$ to a non-empty subset of $\mathbb{L}^d_k$ that is closed under action of $\operatorname{E}(d)$. In concrete, let $A$ be a non-empty subset of $\mathbb{L}^d_k$ satisfies that $g\mathcal{L} \in A$ for all $g \in \operatorname{E}(d)$ and $\mathcal{L} \in A$. Let $f \in L^1(\mathbb{R}^d)$ such that $\mathcal{R}^\alpha_\mathcal{L}f =0$ for all $\mathcal{L} \in A$. Using the same argument, we can demonstrate that $f = 0$. In particular, for $\mathbb{T}$ and $\mathbb{T}^\perp$ are introduced in Subsection 5.2, since both $\mathbb{T}$ and $\mathbb{T}^\perp$ are closed under action of $\operatorname{E}(d)$, we see that a function $f \in L^1(\mathbb{R}^d)$ is equal to $0$, if $\mathcal{R}^\alpha_\mathcal{L}f =0$ for all $\mathcal{L} \in \mathbb{T}$, or for all $\mathcal{L} \in \mathbb{T}^\perp$.
> >
> > **Q2. How does Db-TSW handle cases where the Euclidean distance between points and tree lines is nearly uniform across lines, potentially reducing the distinctiveness of the splitting map?**
> >
> > **Answer.** To compute Db-TSW, as shown in Eq. (21), we use the Monte Carlo method by sampling $L$ tree systems to approximate the distance. As a result, the distances between data points and tree lines vary during training since we sample at each epoch. Therefore, it is not problematic if, for some tree systems, these distances become uniform.
> >
> > ---
> >
> > We sincerely thank the reviewer for the valuable feedback. If our responses adequately address all the concerns raised, we kindly hope the reviewer will consider raising the score of our paper.

---

> ### Author Response · Authors · 2024-11-22
> **Any Questions from Reviewer xP5V on Our Rebuttal?**
>
> We would like to thank the reviewer again for your thoughtful reviews and valuable feedback.
>
> We would appreciate it if you could let us know if our responses have addressed your concerns and whether you still have any other questions about our rebuttal.
>
> We would be happy to do any follow-up discussion or address any additional comments.

---

> > ### Comment · Reviewer_xP5V · 2024-11-25
> >
> > I would like to thank the authors for addressing my concerns in the rebuttal. I'm fairly satisfied with the current state of the paper and, as a result, I would like to raise my score to 8.

---

> ### Author Response · Authors · 2024-11-25
> **Thanks for your endorsement!**
>
> Thank you for your response, and we deeply appreciate your thoughtful endorsement.

---

### Author Response · Authors · 2024-11-21
**General Response (1/3)**

Dear AC and reviewers,

Thanks for your thoughtful reviews and valuable comments, which have helped us improve the paper significantly.

We sincerely thank the reviewers for their valuable feedback and constructive suggestions. We are encouraged by the positive endorsements regarding the following aspects of our work:

1. Paper is very-well written, clear and easy-to-follow. (Reviewer xP5V, 2G1e, vezg, YmcA)

2. The authors propose a novel Radon transform on Systems of Lines, extending the invariant splitting mapping from previous works to project measures on trees, demonstrating its injectivity and introducing the corresponding TSW distance as an invariant metric. Additionally, they present a practical splitting map and a new method for sampling trees. (Reviwer xP5V, vezg, YmcA)

3.  The experiments demonstrate that the proposed distance achieves competitive results, particularly excelling in the challenging high-dimensional generative modeling task with Diffusion, where it outperforms all previous Sliced-Wasserstein variants and the tree-sliced Wasserstein variant (Reviewer xP5V, 2G1e, vezg, YmcA)

---

Below, we address some common points raised in the reviews:

---

> ### Author Response · Authors · 2024-11-21
> **General Response (2/3)**
>
> **Q1. Computation and memory complexity of Db-TSW**
>
> **Answer.** Since GPUs are now standard in machine learning workflows due to their parallel processing capabilities, and Db-TSW is primarily intended use for deep learning tasks like generative modeling and optimal transport problems which are typically GPU-accelerated, we provide both complexity analysis and empirical runtime/memory of Db-TSW with GPU settings. We have also included these information in Appendix C.6 and C.7 of the paper.
>
> Assume $n \ge m$, the computational complexity and memory complexity of Db-TSW are $O(Lknd + Lkn\log n)$ and $O(Lkn + Lkd + nd)$, respectively. In details, the Db-TSW algorithm will have there most costly operations as described in the table below (also Table 5 in Appendix C.6):
>
> *Table: Complexity Analysis of Db-TSW*
> | Operation | Description | Computation | Memory |
> |-----------|-------------|------------|---------|
> | Projection | Matrix multiplication of points and lines | $O(Lknd)$ | $O(Lkd + nd)$ |
> | Distance-based weight splitting | Distance calculation and softmax | $O(Lknd)$ | $O(Lkn + Lkd + nd)$ |
> | Sorting | Sorting projected coordinates | $O(Lkn\log n)$ | $O(Lkn)$ |
> | **Total** | | $O(Lknd + Lkn\log n)$ | $O(Lkn + Lkd + nd)$ |
>
> **The kernel fusion trick**: In the table, the distance-based weight splitting operation involves: (1) finding the distance vector from each point to each line, (2) calculating the distance vectors' norms, and (3) applying softmax over all lines in each tree. The first step costs $O(Lknd)$ computation and $O(Lknd)$ memory. Similarly, the second step costs $O(Lkdn)$ computation and $O(Lknd)$ memory. Lastly, the final step costs $O(Lkn)$ computation and $O(Lkn)$ memory. Therefore, this operation theoretically costs $O(Lknd)$ in terms of both computation and memory. However, we leverage the automatic kernel fusion technique provided by `torch.compile` [(torch document)](https://pytorch.org/docs/stable/generated/torch.compile.html) to "fuse" all these three steps into one single big step, thus contextualizing the distance vectors of size $Lkn \times d$ in a shared GPU memory instead of global memory. In the end, we only need to store two input matrices of size $Lk\times d$ (a line matrix) and $n \times d$ (a support matrix), along with one output matrix of size $Ln \times k$ (a split weight), which helps reduce the memory stored in GPU global memory to only $O(Lkn + Lkd + nd)$.
>
> In Appendix C.7, we have conducted a runtime and memory comparison of Db-TSW with respect to the number of supports and the support's dimension in a single Nvidia A100 GPU. We fix $L=100$ and $k=10$ for all settings and varies $N \in \{500, 1000, 5000, 10000, 50000\}$ and $d \in \{10, 50, 100, 500, 1000\}$.
>
> In Figure 7a (Appendix C.7), the relationship between runtime and number of supports demonstrates predominantly linear scaling, particularly beyond $10000$ supports, though there's subtle non-linear behavior in the early portions of the curves for higher dimensions. The performance gap between high and low-dimensional cases widens as the number of supports increases, with $d=1000$ taking approximately twice the runtime of $d=500$, suggesting a linear relationship between dimension and computational time.
>
> In Figure 7b (Appendix C.7), the memory consumption analysis of Db-TWD reveals several key patterns which align with the theoretical complexity analysis and suggest predictable scaling behavior. First, there is a clear linear relationship between memory usage and the number of samples across all dimensions. Second, in higher-dimensional cases ($d = 100, 500, 1000$), the memory requirements exhibit a proportional relationship with the dimension, as evidenced by the approximately equal spacing between these curves. Finally, for lower dimensions ($d = 10, 50, 100$), the memory curves nearly overlap due to the dominance of the $Lkn$ term in the memory complexity formula $O(Lkn + Lkd + nd)$, where the number of supports components ($Lkn$) has a greater impact than the dimensional components ($Lkd$ and $nd$) when $L=100$ and $k=10$.

---

> ### Author Response · Authors · 2024-11-21
> **General Response (3/3)**
>
> **Q2. The effect of $L$, $k$ and $\delta$ to the method's performance**
>
> **Answer.** In Figure 8 in Appendix C.8, we include the ablation study for $L$, $k$, and $\delta$ on the Gradient flow task. We chose the Gaussian 100d dataset. When not chosen to vary, we used the default values: number of projections $n=100$, learning rate $lr=0.005$, number of iterations $n_{iter}=5000$, number of trees $L=500$, number of lines per tree $k=2$, and coefficient $\delta=10$. We fixed all other values while varying $k\in \{2,4,16,64,128\}$, $L \in \{100, 500, 1000, 1500, 2000, 3000\}$, and $\delta \in \{1, 2, 5, 10, 20, 50, 100\}$.
>
>
> From Figure 8a in Appendix C.8, the ablation analysis of the number of lines $k$ reveals an interesting relationship between the number of lines per tree ($k$) and the algorithm's performance. When fixing the number of trees ($L$), configurations with fewer lines per tree demonstrate faster convergence to lower Wasserstein distances. This behavior is reasonable because increasing $k$ expands the space of tree systems, which in turn requires more samples to attain similar performance levels. For instance, $k=2$ and $k=4$ configurations reach lower Wasserstein distances compared to $k=64$ and $k=128$. Additionally, as demonstrated in the runtime and memory analysis (Appendix C.7), configurations with higher number of lines per tree incur greater computational and memory costs while yielding suboptimal performance.
>
> From Figure 8b in Appendix C.8, the convergence analysis demonstrates a clear relationship between the number of trees ($L$) and the algorithm's performance. When fixing the number of lines per tree ($k$), configurations with more trees achieve significantly better convergence, reaching much lower Wasserstein distances. This behavior is expected because increased sampling in the Monte Carlo method used in Db-TSW leads to a more accurate approximation of the distance. Specifically, comparing $L=3000$ and $L=100$, we observe that $L=3000$ achieves a final Wasserstein distance of $10^{-5}$, while $L=100$ only reaches $10^2$. However, there is a computation-accuracy trade-off since more trees involves more computation and memory.
>
> From Figure 8c in Appendix C.8, the analysis of different $\delta$ values indicates variations in the algorithm's performance. Moderate values of delta ($\delta=1-10$) enhance the convergence speed compared to $\delta = 1$, as they effectively accelerate the optimization process.  However, when delta becomes too large ($\delta=20-100$), the converged speed decreased.
>
> ---
>
> We are glad to answer any further questions you have on our submission.

---

### Author Response · Authors · 2024-11-21
**Summary of Revisions**

Incorporating comments and suggestions from reviewers, as well as some further empirical studies we believe informative, we summarize here the main changes in the revised paper:
1. In **Appendix C.3**, we have added detailed results comparing our method with other baselines on the Gaussian 20d dataset, including mean-standard deviation tables and visualizations to highlight our approach's advantages.
2. We have added **Appendix C.6**. In this section we have provided a detailed analysis of Db-TSW's computational and memory complexity, along with techniques for reducing GPU memory consumption.
3. We have added **Appendix C.7**. In this section, we have included empirical analysis of Db-TSW's runtime and memory usage relative to the number of supports ($N$) and support dimension ($d$).
4. We have added **Appendix C.8**. In this section, we have added an ablation study examining how parameters $L$, $k$, and $\delta$ affect overall performance.
5. We have corrected typos identified by reviewers, including removing a redundant sentence in **Figure 2's caption** (as noted by Reviewer YmcA).

---

### Meta-Review · Area_Chair_TGUm · 2024-12-21

**Metareview:**

This paper introduces Distance-based Tree-Sliced Wasserstein (Db-TSW), an enhancement of Tree-Sliced Wasserstein (TSW-SL) that uses a novel class of splitting maps to incorporate full positional information from input measures. Db-TSW addresses the limitations of TSW-SL by ensuring Euclidean invariance and injectivity via a revised Radon transform. The paper also proposes an efficient GPU-friendly tree sampling process. Experiments demonstrate that Db-TSW achieves competitive accuracy over SW variants while retaining computational efficiency in tasks like gradient flows, image style transfer, and generative modeling.

Major strength: an originall variant of SW, with adequate theoretical background supported by a new Radon transform.

Major weakness: the experimental section may appear weak, or artificial, to fully support the importance of this new variant.

Reviewers mostly agree that the paper present interesting contributions to variants of SW, and that it can yield interesting research paths despite some weaknesses in the experimental section. I am recommending an accept option for this paper.

**Additional Comments On Reviewer Discussion:**

During the rebuttal, several reviewers raised their scores after discussing with the authors.

---

### Decision · Program_Chairs · 2025-01-22

Accept (Poster)